# DOLPHIN: A PROGRAMMABLE FRAMEWORK FOR SCALABLE NEUROSYMBOLIC LEARNING

## ABSTRACT

Neurosymbolic learning has emerged as a promising paradigm to incorporate symbolic reasoning into deep learning models. However, existing frameworks are limited in scalability with respect to both the training data and the complexity of symbolic programs. We propose DOLPHIN, a framework to scale neurosymbolic learning at a fundamental level by mapping both forward chaining and backward gradient propagation in symbolic programs to vectorized computations. For this purpose, DOLPHIN introduces a set of abstractions and primitives directly on top of a high-performance deep learning framework like PyTorch. It thereby enables neurosymbolic programs to be written in a language like Python that is familiar to developers and compile them to computation graphs that are amenable to end-to-end differentiation on GPUs. We evaluate DOLPHIN on a suite of 13 benchmarks across 5 tasks that combine deep learning models for text, image, or video processing with symbolic programs that involve multi-hop reasoning, recursion, and black-box functions like Python `eval()`. DOLPHIN achieves comparable or better accuracy on all benchmarks while taking $0.3\%$-$61.7\%$ of the time (and $23.2\%$ on average) to train these models on the largest input per task compared to baselines Scallop, ISED, and IndeCateR+, which time out on most of these inputs.

## 1 INTRODUCTION

Deep learning has made great strides in tasks such as image classification, speech recognition, and natural language processing. With the emergence of foundation models like GPT-4 and CLIP, deep learning is increasingly applied to more complex tasks. While such models work well for prediction and generation tasks, they are limited in their ability to perform reasoning required for tasks involving structure, logic, and planning, where symbolic approaches traditionally excel (Kambhampati et al.). Neurosymbolic programming (Chaudhuri et al., 2021) has emerged as a promising paradigm to incorporate symbolic reasoning into deep learning models, providing the best of both worlds.

Various frameworks have been developed to improve the programmability and accessibility of neurosymbolic applications (Manhaeve et al., 2018; Li et al., 2023; Solko-Breslin et al., 2024). These frameworks support complex symbolic reasoning features like recursion and black-box functions, implement efficient differentiable reasoning algorithms, and provide bindings for deep learning frameworks like PyTorch. However, these frameworks incur significant overhead during training.

Consider a typical workflow of such a framework in Figure 1(a). We have a supervised learning task with labeled data $(x, y)$, a neural network $M_\theta$ that processes input $x$, and a symbolic program $P_{\texttt{symbolic}}$ that takes the network's output $r$ and produces final output $y$. Existing frameworks, such as Scallop (Li et al., 2023), execute the neural model on GPU but use a separate CPU-based backend (implemented in Rust in Scallop's case) for the symbolic program. Moreover, they introduce inter-process latency in transferring state between the neural and symbolic sub-systems.

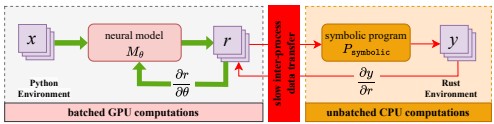
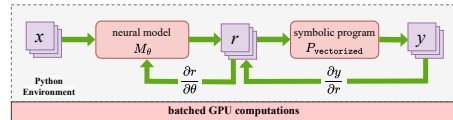

(a) Typical neurosymbolic framework (e.g. Scallop).  (b) Our neurosymbolic framework DOLPHIN.

Figure 1: Comparison of system architectures of neurosymbolic frameworks.

Together, these issues hinder the scalability of neurosymbolic learning with respect to *problem complexity* and *data complexity*. First, the symbolic computation engine must derive a set of all possible results and their associated probabilities in a manner that is differentiable with respect to the network's parameters $\theta$. As the complexity of the symbolic program increases, the number of possible results and their associated weights also grows exponentially, leading to a combinatorial explosion in the number of required computations. However, the symbolic computations are discrete and not easily parallelizable on modern hardware like GPUs. Second, larger datasets also compound the computational cost of neurosymbolic learning. Deep learning typically addresses this challenge by batching computations across multiple data samples. However, in neurosymbolic learning, the computations may differ across data samples, making it difficult to batch them effectively.

To address these challenges, we need to fundamentally rethink the design of a neurosymbolic framework. One approach is to develop specialized and low-level primitives that scale specific benchmarks but make it time-intensive for developers to write neurosymbolic programs tailored to particular tasks. Alternatively, providing high-level primitives—such as a logic programming language like Scallop (Li et al., 2023) or DeepProbLog (Manhaeve et al., 2018)—simplifies the development of symbolic programs but limits the fine-grained control needed to scale specific applications. Finally, to truly democratize neurosymbolic programming, it is crucial to develop a framework that seamlessly integrates into the everyday deep learning workflows that developers already use.

In this work, we propose DOLPHIN, a novel framework for scalable neurosymbolic learning. In DOLPHIN, we build three key components that effectively tackle the scalability and programmability challenges described above. First, we develop a general symbolic representation that efficiently captures the relationships between neural network outputs and associated discrete symbols. Second, we introduce a set of primitives to map forward chaining in symbolic programs to vectorized computations over these representations. Third, we develop a set of vectorized *provenance semirings* (Green et al., 2007) that are easily pluggable into DOLPHIN and enable to efficiently compute symbolic gradients. As illustrated in Figure 1b, these components together allow DOLPHIN to build a computation graph that spans both symbolic and neural operations, is highly parallelizable, and end-to-end differentiable on GPUs. Finally, DOLPHIN is implemented as a library that is integrated with PyTorch, allowing users to easily incorporate it into their existing deep learning pipelines.

We evaluate DOLPHIN on a diverse set of neurosymbolic tasks that involve text, image, video, and multi-modal data, and use rich reasoning features such as recursion and black-box Python functions. Neurosymbolic programs written using DOLPHIN only require 0.3%-61.7% (23.2% on average) of the time to train compared to state-of-the-art baselines including differentiable reasoning frameworks like Scallop, and sampling-based frameworks like ISED and IndeCateR+ while maintaining similar levels of accuracy. We also observe that DOLPHIN efficiently scales to more complex benchmarks and larger datasets whereas the baselines either time out after 10 hours or fail to converge.

We make the following contributions in this work:

- We propose DOLPHIN, a novel neurosymbolic programming framework for end-to-end differentiable symbolic reasoning in a scalable manner.
- We develop novel abstractions to represent symbolic and neural computations and introduce vectorized primitives for neurosymbolic programs.
- We develop vectorized provenances that can be plugged into DOLPHIN for efficient computation of symbolic gradients on parallelizable hardware such as GPUs.
- We evaluate DOLPHIN on a diverse range of challenging neurosymbolic tasks across different domains and show that it effectively scales with increasing problem complexity and dataset size.

## 2 OVERVIEW

We illustrate our approach using the MNIST Sum-$N$ task from (De Smet et al., 2024). The goal is to train a model that takes as input $N$ images of MNIST digits and returns the sum of the digits represented by the images. During learning, supervision is provided only on the sum instead of the labels of the digits. The difficulty of the problem scales exponentially as there are $10^N$ states in the input space. Further, there are only $9N + 1$ possible labels, resulting in very sparse supervision.

Figure 2a shows the code for this task using DOLPHIN with PyTorch. The neural module is a convolutional neural network (CNN) called `MNISTNet`. It takes in a batch of image tuples `imgs` where each sample contains $N$ MNIST images. `MNISTNet` classifies each image into one of 10

```
1  class MNISTNet(nn.Module):
2    def __init__(self):
3      super(MNISTNet, self).__init__()
4      ...
5
6  class SumNNet(nn.Module):
7    def __init__(self):
8      super(SumNNet, self).__init__()
9      self.CNN = MNISTNet()
10
11   def forward(self, imgs):
12     digits = range(10)
13     D_res = Distribution(self.CNN(imgs[0]), digits)
14     for i in range(1, len(imgs)):
15       D_i = Distribution(self.CNN(imgs[i]), digits)
16       D_res = apply(D_res, D_i, lambda x,y: x + y)
17     l_res = get_logits(D_res)
18     return l_res
```

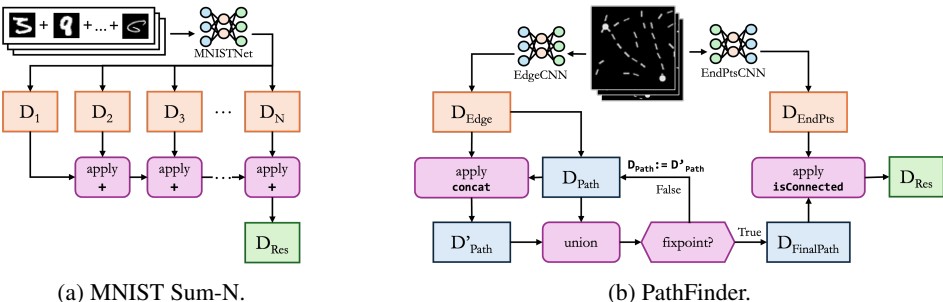

(a) Code using DOLPHIN primitives with PyTorch.   (b) Components of the SumNNet model.

Figure 2: A neurosymbolic program for the MNIST Sum-N task written using DOLPHIN.

(a) MNIST Sum-N.          (b) PathFinder.

Figure 3: Computation graphs for two neurosymbolic programs written using DOLPHIN.

classes representing the digits 0-9. The logits produced by MNISTNet, representing probability distributions over the digits, are then passed as inputs to the symbolic program. Lines 13-17 depict a symbolic program written in Python using DOLPHIN primitives.

In order to support training, the symbolic program must reason over all the outputs of the CNN, and return probability distributions over all the possible results ($0$ to $9N$). This involves tracking the probabilities of individual *symbols* (here, digits or numbers), combinatorially evaluating the results of complex symbolic functions, and calculating the probabilities of each intermediate result, all while tracking their gradients to allow for accurate backpropagation while optimizing the training objective. The batched nature of data in machine learning further complicates these calculations since the probabilities of symbols can be different across samples within the same batch. As a result, writing neurosymbolic programs in native PyTorch is tedious even for simple tasks.

To address these issues, DOLPHIN provides primitives that allow programmers to express symbolic programs without worrying about the underlying computations. Lines 13 and 15 of Figure 2a show how the CNN's output can be captured within Distribution objects. Each Distribution associates the digits with the corresponding batched logits produced by the CNN, along with any gradients and associated metadata. Figure 2b shows the internal structure of these objects.

The programmer can now express the symbolic program in terms of operations that manipulate Distributions. For instance, in line 16, the apply function is used to perform an operation on two distributions. Here, the apply function takes two Distributions as arguments, along with a lambda function that specifies the addition operation. Under the hood, apply combinatorially explores all the possible sums of the symbols from D_res and D_i and calculates their associated probabilities using an appropriate provenance. The result of apply is a new Distribution over the calculated sums, and is stored back into D_res. This is repeated iteratively until all the outputs of the CNN are summed appropriately. Once the final Distribution is calculated, it is simply a matter of getting its logits which can be used to calculate the loss of the predictions.

DOLPHIN provides additional primitives to support more complex symbolic programs. Figure 3b, for instance, shows the computation graph for the PathFinder task (Tay et al., 2021), which involves

$$
\begin{array}{rcl}
\text{Symbol} &::& s \ \in \ S \ \text{(objects)} \\
\text{Tag} &::& t \ \in \ T \ \text{(tensors)} \\
\text{Distribution} &::& D \ \in \ \mathbb{D} = S \to T
\end{array}
\qquad
\begin{array}{rcl}
\text{APPLY} &:& \mathbb{D}^K \times (S^K \to S) \to \mathbb{D} \\
\text{FILTER} &:& \mathbb{D} \times (S \to \mathbb{B}) \to \mathbb{D} \\
\text{APPLYIF} &:& \mathbb{D}^K \times (S^K \to S) \times (S^K \to \mathbb{B}) \to \mathbb{D} \\
\text{UNION} &:& \mathbb{D} \times \mathbb{D} \to \mathbb{D} \\
\text{GETPROBS} &:& \mathbb{D} \to [0, 1]^N
\end{array}
$$

Figure 4: Formal definition of DOLPHIN's programming abstractions and primitives.

recursively building paths to identify if two points in a maze are connected. The `union` primitive is used to support the recursive nature of this program. As with `apply`, DOLPHIN maps the symbolic operations denoted within these primitives to probability computations. Given that `Distribution` objects associate symbols with the batched logits themselves, these computations are vectorized and directly operate over PyTorch tensors. This deep integration of DOLPHIN into PyTorch allows programmers to write symbolic programs as *symbolic layers* that interact with standard PyTorch neural layers within a neurosymbolic model. DOLPHIN can thus leverage the hardware acceleration supported by PyTorch to scale to large and complex programs. This contrasts with systems like Scallop (Li et al., 2023), where tensors are converted into Scallop-friendly tags transferred to a process outside the Python environment with CPU-bound probability computations, restricting scalability.

# 3 THE DOLPHIN FRAMEWORK

## 3.1 DOLPHIN CORE DESIGN PRINCIPLES

We based DOLPHIN's framework design on the following core principles:

- **Flexible programmability:** The framework should allow developers to write neurosymbolic applications in Python with minimal effort, providing intuitive primitives that seamlessly integrate with Python's rich and expressive language features.
- **End-to-end differentiability on GPUs:** The framework should allow any neurosymbolic program to be end-to-end differentiable on GPUs irrespective of the task characteristics.
- **Scalable:** The framework should easily scale with greater problem and data complexity.
- **Tunable:** Similar to hyperparameters in deep learning, the framework should provide a simple interface for developers to choose provenances (and their configurations) or define new ones.

Together, these principles help address the challenges of scaling neurosymbolic frameworks. The flexible programmability and tunability allow us to write complex neurosymbolic programs, while GPU differentiability and scalability work towards addressing data complexity. We show how we realize these principles by describing the key components of DOLPHIN.

## 3.2 THE DOLPHIN SYNTAX

DOLPHIN provides a programming interface that developers can use to express symbolic programs in a Pythonic manner. DOLPHIN maps each operation of the symbolic program to PyTorch which enables end-to-end GPU-accelerated differentiable reasoning. Figure 4 presents DOLPHIN's programming interface including the symbolic abstractions and operations over them.

### 3.2.1 ABSTRACTIONS

The three main abstractions provided by DOLPHIN for expressing differentiable symbolic programs are shown on the left of Figure 4. *Symbols* $S$ represent symbolic entities relevant to the program. These entities can be any Pythonic object, such as hand-written digits in MNIST-SumN or coordinates of points in PathFinder. *Tags* $T$ are tensors that represent their likelihoods. Typically, tags for symbols are derived from the outputs of machine learning models, such as the probability distribution over digits produced by the CNN classifier in MNIST-SumN. Finally, *Distribution* $D$ represents the likelihood of an input being classified as one of the pre-defined symbols.

Distributions serve as the fundamental datatype of a DOLPHIN program and act as its main interface with a machine learning model. For instance, when the developer instantiates a Distribution object, such as in the following code snippet from Figure 2a:

```
D_res = Distribution(self.CNN(imgs[0]), digits)
```

the output of the CNN model is directly passed to the Distribution object, effectively acting as an input to the symbolic program. The Distribution object itself, as shown in Figure 2b, contains batches of *tags* extracted from the model outputs, and maintains the set of corresponding symbols.

To enable such a seamless integration between the PyTorch model and the symbolic program, Distributions are designed to operate directly over PyTorch tensors. This has two main advantages. First, it preserves the gradients of the model output throughout the symbolic program, enabling end-to-end differentiability. Second, it allows DOLPHIN to perform operations over an entire batch of tags, leveraging the vectorized operations provided by PyTorch. DOLPHIN can thus operate efficiently on specialized hardware like GPUs, allowing the symbolic program to scale effectively.

### 3.2.2 OPERATIONS

Figure 4 shows the five operations supported by DOLPHIN that developers can use to manipulate Distributions and express complex symbolic programs. We now expand on these operations.

**APPLY.** This is the primary operation that developers can use to manipulate Distributions. It takes as inputs $K$ Distributions, where $K \geq 1$, along with a function $f$ of the same arity. This function defines operations over the symbols of $K$ distributions. APPLY then computes the results of $f$ over all possible combinations of arguments sourced from the symbols of the Distributions as well as their associated tags, and returns a new Distribution with these results and tags.

This operation occurs in two stages akin to the popular map-reduce pattern. In the *map* stage, APPLY computes the results of $f$ over the symbols of the input Distributions and their associated tags:

$$R = \{ (f(s_1, s_2, \ldots, s_k), (t_1 \otimes t_2 \otimes \ldots \otimes t_k)) \mid D_i(s_i) = t_i, i = 1, \ldots, k \} \tag{1}$$

Here, the tag of each result symbol $f(s_1, s_2, \ldots, s_k)$ is the conjunction $\otimes$ of the tags $(t_1, t_2, \ldots, t_k)$ of the input symbols it was derived from. While the tag computations are performed on the GPU, the function $f$ is executed sequentially on the CPU for each combination of symbols. This is because function $f$ can be any user-defined Python function, including complex control flows and operations like regex parsing, image processing, or Python's `eval()`. It may also be a many-to-one function and the tags shared by a resulting symbol must be aggregated to form the final tags of the output Distribution. We, therefore, *shuffle* the results from the map stage to compute a function $M$ from each symbol to tags from $R$ associated with it:

$$M = \lambda s \, . \, \{ t \mid (s, t) \in R \} \tag{2}$$

We then proceed to the *reduce* stage, where we aggregate the tags of each symbol in $M$ using disjunction $\oplus$ to produce the final Distribution $D_{\text{res}}$:

$$D_{\text{res}} = \lambda s \, . \, \bigoplus \{ t \mid t \in M(s) \} \tag{3}$$

Since the tags here are PyTorch tensors representing probabilities, the implementations of the conjunction and disjunction operations are dictated by the underlying provenance used by the program. A more detailed explanation of the provenances is provided in Section 3.4.

**FILTER.** The FILTER operation is used to filter out symbols from a Distribution based on some condition. It takes in a single Distribution, along with a user-defined function that returns a boolean value, which acts as the condition. This operation then returns a new Distribution that contains only the symbols that satisfy the condition, along with their tags.

**APPLYIF.** This operation is a conditional version of APPLY. It takes in $K$ Distributions and functions $f_{apply}$ and $f_{cond}$ of the same arity. For each combination of symbols from the $K$ Distributions, APPLYIF computes $f_{apply}$ and its associated tags only if the condition $f_{cond}$ is satisfied over that combination of symbols. The operation then returns a new Distribution with these results and tags.

**UNION.** The UNION operation is used to combine two Distributions. It takes in two Distributions and returns a new Distribution that contains the union of the symbols from the two input Distributions, along with their tags. Any symbols common to both input Distributions have their tags merged via a disjunction operation. UNION is especially useful when writing recursive programs in DOLPHIN that require combining the results of multiple recursive calls, as described in Appendix D.

**GETPROBS.** The GETPROBS operation extracts the probabilities from the tags of a Distribution. This is used mainly once the symbolic program has been executed to extract the final probabilities

| Provenance | Domain | 0 | 1 | $t \oplus t'$ | $t \otimes t'$ |
|---|---|---|---|---|---|
| DAMP | $[0, 1]$ | $0$ | $1$ | $\text{clamp}_0^1(t + t')$ | $t \cdot t'$ |
| DTKP-AM | $[0, 1] \cup \{\infty, -\infty\}$ | $\hat{\mathbf{0}}_{ij} = -\infty$ | $\hat{\mathbf{1}}_{ij} = \begin{cases} \infty & i = 1 \\ -\infty & i > 1 \end{cases}$ | $\text{top}_k(\text{cat}(t, t'))$ | $\text{top}_k([\min(|t_i|, |t_j'|) \mid (t_i, t_j') \in t \times t'])$ |

Table 1: DOLPHIN provenances implemented in PyTorch.

of the symbols in the output Distribution. These probabilities can then be used to compute the loss function for training the neural model. The actual extraction of the probabilities from the tags depends on the specific provenance used in the program.

### 3.3 CONTROL FLOW AND RECURSION

Expressing control flow and recursion in a DOLPHIN program can be done in one of two ways. The simplest way is to specify any control flow operations within the user-defined functions supplied to APPLY, APPLYIF, and FILTER, since these functions can contain arbitrary Python code.

Alternatively, one can specify control flow and recursion outside of these functions by specifying all branches separately and merging their results using UNION. Figure 5 shows an example of transitive closure in DOLPHIN, where the `compute_paths` function computes the transitive closure of the graph by iteratively applying edges predicted by a neural model to paths. The APPLYIF function applies the edges to the paths if the end of the first path is the same as the start of the second path. The UNION function merges the new paths with the existing paths. The function `compute_paths` is called recursively until a fixpoint is reached, specifically until no new paths can be added.

```python
def compute_paths(paths, edges):
    new_paths = apply_if(paths, edges, \
        lambda p1, p2: (p1[0], p2[1]), \
        lambda p1, p2: p1[1] == p2[0])
    merged = union(paths, new_paths)
    # checking for convergence via fix-
      point
    if merged.symbols == paths.symbols:
        return merged
    else:
        return compute_paths(merged, edges)

edges = Distribution(model(img), points)
paths = compute_paths(edges, edges)
```

Figure 5: Transitive Closure in DOLPHIN.

### 3.4 DOLPHIN PROVENANCES

As discussed earlier, each symbol in a distribution is associated with a batch of one or more tags. The DOLPHIN primitives define how to manipulate certain tags. For instance, Equations (1) and (3) specify the tags to be conjuncted or disjuncted. We now define the semantics of these operations.

The goal of such operations is to approximate the probabilities of the symbols in the output distribution as accurately as possible. This is achieved by using a mathematical framework called *provenance semirings* (Green et al., 2007). Provenance semirings provide generalized algebraic structure to propagate tags when computing over tagged data. In the case of DOLPHIN distributions, we can view the tags as representing the probabilities, and the data as the distribution's symbols.

Designing and implementing provenances can be challenging since they must be accurate enough to capture the semantics of the symbolic program, while at the same time being coarse enough to maintain computational feasibility. Furthermore, the provenances must be differentiable to enable end-to-end training for neurosymbolic tasks. While neurosymbolic frameworks like Scallop (Li et al., 2023) implement differentiable provenances, they are not designed to leverage hardware accelerations or batched optimizations due to the CPU-bound nature of their implementations. We thus design vectorized provenances in DOLPHIN that are differentiable and enable GPU computations.

We simplify the definition of provenances from Scallop as a 5-tuple: $(T, \mathbf{0}, \mathbf{1}, \otimes, \oplus)$. Here, $T$ is the tag space, $\otimes : T \times T \to T$ is the conjunction operator with identity $\mathbf{0}$, and $\oplus : T \times T \to T$ is the disjunction operator with identity $\mathbf{1}$. We then implement two differentiable provenances in DOLPHIN: Differentiable Add-Mult Probabilities (DAMP) and Differentiable Top-K Proofs (DTKP). Table 1 summarizes the operations of these provenances.

**Differentiable Add-Mult Probabilities.** Differentiable Add-Mult Probabilities (DAMP) is a popular technique that uses the probability space as its tag space: $T = [0, 1]$. Its conjunction operation $\otimes$ is defined as the product of probabilities, clamped at $\mathbf{1}$, and its disjunction operation $\oplus$ is defined as the sum of probabilities. The main assumption underlying the DAMP operations is that the input

Distributions are mutually exclusive and independent. This assumption allows DAMP to compute probabilities extremely efficiently, as the operations are simple and can be easily vectorized.

**Differentiable Top-$k$ Proofs.** Differentiable Top-$k$ Proofs (DTKP) (Huang et al., 2021) was proposed to overcome the shortcomings of DAMP. This provenance tracks a set of up to $k$ *proofs* for each symbol. Each proof, in turn, denotes the set of input symbols necessary to derive the output symbol. These proofs are then used to compute the probabilities of the output symbols. In Scallop, DTKP tags are converted into probabilities via differentiable weighted model counting (WMC). This form of DTKP, which we call DTKP-WMC, is computationally hard and is by nature difficult to vectorize due to the varying sizes of proof sets and the WMC procedure. We hence design a vectorized approximation of DTKP-WMC, called DTKP-AM (DTKP with Add-Mult), that can be efficiently computed on GPUs.

We first define the structure of tags in DTKP-AM in a manner that conforms to the constraints of PyTorch tensors. Each tag $t$ for a symbol $s$ is a 2-dimensional tensor of shape $(k, |I|)$, where $k$ is the maximum number of proofs to be retained and $I$ is an ordered list of all *input symbols* (symbols that are present in the input Distributions). Each row $t_i$ of $t$ corresponds to one of the tag's $k$ proofs. Each element $t_{ij}$ thus represents the probability of the $j$th input symbol in the $i$th proof:

$$t_{ij} = \begin{cases} p_j & \text{if the } j\text{th symbol is present in the } i\text{th proof} \\ \hat{\mathbf{0}}_{ij} & \text{otherwise} \end{cases}$$

where $p_j$ is the probability of the $j$th input symbol. The probability of each proof is then computed by taking the product of the normal:

$$\Pr(t_i) = \prod_j \text{norm}(t_{ij}) \quad \text{where} \quad \text{norm}(t_{ij}) = \begin{cases} 1 & t_{ij} = +\infty \\ 0 & t_{ij} = -\infty \\ t_{ij} & \text{otherwise} \end{cases}$$

We next define the operations of DTKP-AM in Table 1. The $\oplus$ operation is defined as the union of two tag tensors $t$ and $t'$ while $\otimes$ is defined as the element-wise minimum of the normalized elements of all possible combinations of proofs in $t$ and $t'$. In each case, the top$_k$ operation retains only upto $k$ proofs with the highest probabilities. These definitions thus allow us to take advantage of the benefits of the DTKP provenance while enabling efficient computation on GPUs. To calculate the probability of the entire tag, DTKP-AM adds the probabilities of the individual proofs and clamps it at 1. We provide a detailed discussion of DTKP-AM in Appendix C.

### 3.5 BUILDING THE DOLPHIN PROGRAM

The programmer specifies the neurosymbolic task using a Python program $P$ that uses DOLPHIN's programming interface to connect the neural components (e.g., neural networks) with the symbolic components and operations. We call $P$ the *symbolic program*. Because $P$ is a Python program and DOLPHIN interfaces with PyTorch, DOLPHIN supports any PyTorch-based neural network(s), most Python language features, and custom user-defined functions. This feature enables greater flexibility and expressiveness in neurosymbolic programs than existing frameworks.

In addition to $P$, the programmer provides one or more neural networks $M_1, \ldots, M_k$, and a dataset $\mathcal{D}$. Given these inputs, DOLPHIN extracts the computation graph that encodes how the neural network outputs are transformed using symbolic operations to produce a final result $D_{\text{res}}$. All computations in DOLPHIN are expressed using distribution objects $D_i$. Each DOLPHIN primitive (e.g., APPLY) takes one or more distribution objects as inputs and applies a transformation to produce another distribution object. Because each distribution object $D_i$ only contains vectors of tags (or probabilities), the entire computation graph (including the neural network(s)) can be ported to a GPU for faster execution. During training, DOLPHIN optimizes over the standard objective function:

$$\phi(\theta) = \min_\theta \sum_{(x,y) \in \mathcal{D}} \mathcal{L}(P(M_\theta(x)), y) \tag{4}$$

Here $\mathcal{L}$ is the loss function, such as binary cross entropy. While DOLPHIN allows $P$ to take multiple neural networks as inputs, we show only one neural network model $M$ here for simplicity.

### 4 EXPERIMENTS

We evaluate DOLPHIN on a set of 13 benchmarks of varying complexity and scale across 5 neurosymbolic tasks. Our evaluation addresses the following research questions:

- **RQ1: Scalability.** How does DOLPHIN scale to complex problems and large datasets?
- **RQ2: Accuracy.** Do models written in DOLPHIN converge to SOTA accuracies?
- **RQ3: Provenance Comparisons.** Which provenances are most effective for each benchmark?

## 4.1 BENCHMARKS

We evaluate DOLPHIN on the following benchmarks. We give additional context and information about the experiment setup for each branch in Appendix A.

**MNIST-SumN.** The MNIST-SumN (or briefly, SumN) task from (De Smet et al., 2024) takes as inputs $N$ handwritten digits from the MNIST dataset and returns their sum. We consider three versions of this task: small ($N = 5$), medium ($N = 10$), and large ($N = 15$).

**Hand-Written Formula (HWF).** The HWF task from Li et al. (2020) takes as input a set of images of handwritten digits and arithmetic operators representing a formula. The task is to evaluate the formula and return the result. We consider three versions of HWF: small (formulas of length up to 7), medium (formulas of length up to 15), and large (formulas of length up to 19).

**PathFinder.** PathFinder (or Path) from Tay et al. (2021) tests the ability of an agent to reason over long-range dependencies within an input image. The image consisting of two dots and a sequence of curved and dashed lines. The task is to identify whether the two dots are connected via the lines. We consider three versions of this task based on the image size in pixels: small (32 x 32), medium (128 x 128), and large (256 x 256).

**CLUTRR.** In this task from Sinha et al. (2019), the input is a passage of text containing some information about several individuals and some of their relationships. The task is then to infer the relationship between two given individuals, which is not explicitly provided in the input. We consider two versions of this task, where the training data contains relation chains of lengths up to 3 (small) or 4 (medium).

**Mugen.** In this task from Hayes et al. (2022), the input is a video of gameplay footage that is 3.2 seconds long and a natural language passage captioning the video. The goal is to measure how aligned the text is with the video. This task has two variants: Mugen-TVR, where the model must retrieve the video that best aligns with the text, and Mugen-VTR, where the model must retrieve the text that best aligns with the video. This benchmark tests the ability of the model to reason over multimodal data. We consider two versions of this task: small, with 1000 training samples, and medium, with 5000 training samples.

## 4.2 EXPERIMENTAL SETUP AND BASELINES

**Setup.** All experiments, except for the CLUTRR benchmark, were run on machines with two 20-core Intel Xeon Gold 6248 CPUs, four NVIDIA GeForce RTX 2080 Ti GPUs, and 768 GB RAM. Since the CLUTRR benchmark requires more GPU memory, it was run on a machine with 8 NVIDIA A100 40GB GPUs instead. We ran each tool thrice until convergence or until a timeout of 10 hours was reached and report the average best accuracy and training time. The code of DOLPHIN and the benchmarks are provided in the supplementary material.

**Baselines.** We select Scallop (Li et al., 2023), a contemporary state-of-the-art neurosymbolic framework supporting differentiable programming optimized to run on the CPU and use multiple cores to parallelize its computations. We also choose two sampling-based gradient approximation methods, ISED (Solko-Breslin et al., 2024) and IndeCateR+ (De Smet et al., 2024). We compare DOLPHIN against Scallop on all benchmarks, and against ISED and IndeCateR on MNIST-SumN and HWF.

## 4.3 RQ1: SCALABILITY

Table 2 presents the total training times ($T_{total}$) in seconds for DOLPHIN and baselines on all benchmarks, along with the time per epoch ($T_{epoch}$) and scaling factor $\alpha$. The scaling factor is the ratio of the per epoch times of the baselines to DOLPHIN. We observe that in almost all cases, DOLPHIN shows significant improvements in training times, training up to about 300x faster than Scallop, 44x faster than ISED, and 3x faster than IndeCateR+. On average, DOLPHIN reports a speedup of 21x times over all baselines. Further, DOLPHIN converges on all benchmarks, while the baselines time

| Task | DOLPHIN | | Scallop | | | ISED | | | IndeCateR+ | | |
|------|---------|---------|---------|---------|------|---------|---------|------|---------|---------|------|
| | $T_{total}$ | $T_{epoch}$ | $T_{total}$ | $T_{epoch}$ | $\alpha$ | $T_{total}$ | $T_{epoch}$ | $\alpha$ | $T_{total}$ | $T_{epoch}$ | $\alpha$ |
| SumN (S) | 78.33 | **15.67** | 923.78 | 184.76 | 11.80 | 299.63 | 59.93 | 3.82 | 416.78 | 59.54 | 3.8 |
| SumN (M) | 144.92 | **14.49** | 3.41e3 | 341.57 | 23.57 | 2.16e3 | 216.54 | 14.94 | 385.65 | 32.14 | 2.22 |
| SumN (L) | 220.47 | **14.70** | 7.41e3 | 493.87 | 33.60 | 9.8e3 | 653.39 | 44.45 | 548.28 | 23.84 | 1.62 |
| HWF (S) | 3.17e3 | 158.79 | 9.99e3 | 499.57 | 3.15 | 1.58e4 | 790.04 | 4.97 | 1.35e4 | 540.26 | 3.4 |
| HWF (M) | 1.46e4 | **731.67** | TO | 1.49e4 | 20.41 | TO | 6.83e3 | 9.34 | 2.51e4 | 2512.03 | 3.43 |
| HWF (L) | 2.42e4 | **1.21e3** | TO | 3.92e5 | 323.21 | TO | 1.05e4 | 8.61 | TO | 4091.46 | 3.37 |
| Path (S) | 1.08e4 | **1.08e3** | 2.2e4 | 2.2e3 | 2.03 | | N.A. | | | | |
| Path (M) | 1.79e4 | **1.79e3** | TO | 4.17e3 | 2.32 | | | | | | |
| Path (L) | 1.94e4 | **1.94e3** | TO | 1.12e4 | 5.81 | | | | | | |
| CLUTRR (S) | 1.54e3 | **154.85** | 4.29e3 | 429.97 | 2.77 | | N.A. | | | | |
| CLUTRR (M) | 2.91e3 | **291.36** | 7.83e3 | 783.11 | 2.69 | | | | | | |
| Mugen (S) | 3.62e3 | 180.80 | 1.34e4 | **133.68** | 0.74 | | N.A. | | | | |
| Mugen (M) | 1.78e3 | 890.34 | TO | **634.86** | 0.71 | | | | | | |

Table 2: Comparison of training times taken by each baseline. The Timeout (TO) is set at 10 hours. $\alpha$ is the scaling factor, which is the ratio of the per epoch training times of the baselines and DOLPHIN.

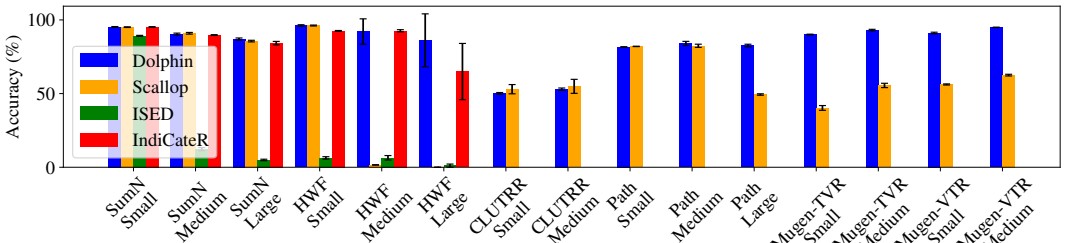

Figure 6: Accuracy of DOLPHIN and baselines across all benchmarks.

out on most of the Medium and Large versions of the benchmarks. These results thus demonstrate that, unlike existing tools, DOLPHIN can scale to complex problems and large datasets.

DOLPHIN trains slightly slower than Scallop on both versions of the Mugen benchmark. This is because the DOLPHIN program written for Mugen uses Python objects and operations that are not fully batchable across samples. In contrast, the Scallop program, which is written in a compiled and optimized language, runs around 1.3x faster than DOLPHIN on average per iteration. However, DOLPHIN requires only 20 epochs to converge whereas Scallop requires almost 1000 epochs (Li et al., 2023). As a result, DOLPHIN's total training time is still significantly lower than Scallop's (3.7x for Mugen(S)). We show the accuracy curves for Mugen in Appendix A.6.

### 4.4 RQ2: ACCURACY

Figure 6 presents the accuracy of DOLPHIN and the baselines on the different benchmarks. DOLPHIN accuracies are marked in blue. In all cases, for DOLPHIN, we report the accuracies of the best-performing provenance. We use the DAMP provenance for MNIST, CLUTRR, and Mugen benchmarks, and the DTKP-AM provenance for the HWF and PathFinder benchmarks.

We observe that in all cases, DOLPHIN achieves state-of-the-art accuracy among neurosymbolic frameworks, except in CLUTRR, where DOLPHIN's accuracy is slightly lower than Scallop's. While DeepProbLog (Manhaeve et al., 2018) reports near-perfect accuracies for CLUTRR, they use negative mining techniques to provide additional labels at train time. Scallop and DOLPHIN, on the other hand, stick to a traditional semi-supervised multiclass classification approach. For most Medium and Large versions of the benchmarks, DOLPHIN achieves better accuracy, whereas the baselines either report lower accuracy due to the complexity of the benchmark (e.g., HWF) or fail to converge within the time limit (e.g., Scallop on PathFinder-Large). Most importantly, these results show that DOLPHIN's scalability improvements do not come at the cost of accuracy.

### 4.5 RQ3: PROVENANCE COMPARISONS

We perform ablation studies to compare the effectiveness of the DAMP and DTKP-AM provenances for each benchmark. We share the graphs in Figure 9 (Appendix B). In all cases, training with the DAMP provenance takes around 132.96 seconds per epoch less than with DTKP-AM on average.

However, the effectiveness of each provenance varies from benchmark to benchmark. For all variations of CLUTRR, Path, and Mugen, both provenances achieve comparable accuracies, with DTKP-AM usually having a slight edge. In the MNIST-SumN benchmark, the DAMP provenance is more effective than the DTKP-AM provenance by 72.08 %pts on average, since the top-k proofs cannot capture all the possible ways in which the sum of the digits can be computed.

In contrast, for HWF, the DTKP-AM provenance is more effective than DAMP by an average of 42.18 %pts. Each step of the HWF program, shown in Appendix G, involves both a concatenation operation and a partial parsing operation before the final expression is evaluated to produce a result. As such, it is difficult for the tags in DAMP to capture the semantics of the symbolic program. In the case of DTKP-AM, each tag is a collection of proofs over input symbols corresponding to logits derived from the neural model. Therefore, any calculated gradients can be directly backpropagated to the logits that most influenced the output, making this a more effective provenance for this task.

## 5 RELATED WORK

**Neurosymbolic programming frameworks.** Frameworks like Scallop (Li et al., 2023), Deep-ProbLog (Manhaeve et al., 2018), and ISED (Solko-Breslin et al., 2024) provide a simple interface for neurosymbolic programming. There are also domain-specific tools like NeurASP Yang et al. (2021) for answer set programming and NeuralLog Chen et al. (2021) for phrase alignment in NLP. While these frameworks provide intuitive abstractions, they are bottle-necked due to expensive data transfers between symbolic computations done on CPU versus neural computations that execute on GPU, making neurosymbolic learning hard to scale. In contrast, DOLPHIN provides a deeper integration of the two worlds by building a Python-based API on top of PyTorch, which scales better.

**Scaling techniques.** Several optimization techniques have been proposed to improve the scalability of differentiable reasoning algorithms. Some techniques aim to scale reasoning algorithms by compiling the symbolic program into computation graphs that can be run on GPUs. LYRICS (Marra et al., 2019), Logic Tensor Networks (Badreddine et al., 2022), and Tensorlog (Cohen et al., 2020) are examples of such techniques. However, these methods focus on first order logic programs and provide limited support for user-defined Pythonic functions, essential for building complex neurosymbolic programs.[1] Greedy NTP (Minervini et al., 2020a) reduces the computation cost of NTP (Rocktäschel & Riedel, 2017) by tracking only a subset of proof states using nearest neighbor search. Likewise, the conditional theorem prover (Minervini et al., 2020b) employs a machine learning-based proof selection technique. However, unlike DOLPHIN, these methods are point solutions that do not fundamentally address the scalability challenge for neurosymbolic learning.

**Specialized neurosymbolic solutions.** There are many specialized solutions for various neurosymbolic tasks. For instance, NGS (Li et al., 2020) uses a hand-coded syntax to specify the structure of mathematical expressions for HWF. More general solutions, such as NS-CL (Mao et al., 2019) includes a framework for visual question answering that learns symbolic representations for text and images. NeRd (Chen et al., 2021) transforms questions in natural language into executable programs based on symbolic information extracted from text. Orvieto et al. (2023) proposes a recurrent neural network architecture that achieves 95% accuracy on Path (S) and 94% on Path (M). In contrast, Dolphin is a general programming framework that tries to scale diverse neurosymbolic programs.

## 6 CONCLUSION AND LIMITATIONS

We proposed DOLPHIN, a programmable framework for scaling neurosymbolic learning. DOLPHIN provides abstractions for writing symbolic programs along with pluggable vectorized provenances to compute symbolic gradients. This allows users to write differentiable symbolic programs in Python within PyTorch pipelines that can scale to complex programs and large datasets. We show that DOLPHIN scales significantly better than existing neurosymbolic frameworks while achieving state-of-the-art performance on a variety of tasks.

A limitation of DOLPHIN is that it needs the user to write programs in a batched manner. While this is a common pattern within deep learning, it may be restrictive to users new to batched programming. Also, while DOLPHIN works well with most models, the representation needed by generative models (e.g., Causal LLMs like Llama) has not been investigated. A third limitation is that Dolphin lacks support for non-deterministic symbolic programs. We leave this to future work.

---

[1]Refer to Appendix E for a detailed comparison of DOLPHIN with LTN.

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
