# OpenReview forum: "Dolphin: A Programmable Framework for Scalable Neurosymbolic Learning"
_ICLR.cc/2025/Conference — Submitted to ICLR 2025_

### Official Review · Reviewer_1NAS · 2024-10-17

**Soundness:** 1
**Presentation:** 3
**Contribution:** 2
**Rating:** 3
**Confidence:** 5

**Summary:**

Many neurosymbolic frameworks have been introduced in recent years, which typically execute their symbolic component on the CPU. This limits their scalability, due to the inferior CPU performance and data transfer latency between the CPU and GPU. The paper introduces Dolphin, a neurosymbolic framework that is fully implemented by parallel tensor operations, and hence can be run on the GPU using conventional deep learning libraries such as PyTorch. The experiments indicate that Dolphin exhibits considerable speed-ups compared to existing frameworks such as Scallop.

**Strengths:**

- The problem tackled in the paper (scalable and efficient neurosymbolic learning) is relevant and an important issue for the broader neurosymbolic community.
- Overall, I feel that the paper is rather well-written and includes several useful figures and examples, resulting in a clear presentation of the ideas.
- The design of Dolphin seems more accessible to a deep learning audience compared to many existing neurosymbolic systems. Notably, it does not require knowledge of e.g. ASP or Prolog (as is the case for NeurASP or DeepProbLog) and can be integrated easily into a deep learning library such as PyTorch or Tensorflow.

**Weaknesses:**

**Novelty**. The key contribution of the paper - speeding up a neurosymbolic framework by tensorizing it and running on GPUs - is certainly not a new idea. One example of prior work is the LYRICS framework by Marra et al. (2019), which also uses tensor operations to perform the symbolic operations in parallel on the GPU (see e.g. Figure 1 in their paper). Logic tensor networks (Badreddine et al., 2022) and Tensorlog (Cohen, 2020) are some additional examples. These frameworks often also support different provenances / semirings / t-norms. The parallelization of neurosymbolic methods with expressive probabilistic semantics is more challenging, but also here there is plenty of existing work (see e.g. Darwiche (2020) or Dang et al. (2021)). Unfortunately, the paper does not mention prior work on parallelized neurosymbolic learning, nor how it is different from these existing methods.

**Semantics**. It is not clear to me what exact semantics Dolphin aims to achieve. The first provenance (DAMP) is essentially fuzzy semantics (which already has been shown to be easily parallelizable, e.g. Badreddine et al. (2022)). On the other hand, “Apply” mostly brute-force enumerates models meaning the necessary independence assumptions for probabilistic sematics can often hold. (c.f. the MNIST-experiment). The second provenance is the top-k semiring, which is less trivial to parallelize. However, the proposed solution of adding the different proofs destroys the top-k semantics (lower-bounding the WMC). This also results in the introduction of clamp operations, which could lead to zero gradients.

**Language**. A distinction from existing methods is that Dolphin introduces its own set of programming primitives (apply, filter, etc.). Previous neurosymbolic frameworks have typically built on an existing language, e.g. Datalog for Scallop, ASP for NeurASP, ProbLog for DeepProbLog, etc. However, there is no justification for the choice of programming primitives. How does its expressivity relate to existing systems such as Scallop? Why wasn’t an existing language chosen? In my opinion, a lot of different choices could have been made (e.g. why do you need ApplyIf instead of just Apply + Filter?).

**Experiments**. I was surprised that the IndeCateR baseline achieved such low accuracy, given that the experiment seems to be the same as in the IndeCater paper, where the reported results are much better. I just tried out the original IndeCateR implementation myself, and I could replicate the MNIST-addition (L) on my machine in 2 minutes. In contrast, the paper reports a timeout after 10 hours. The accuracy also reaches 86.8%, as opposed to less than 10% in the paper (I'm not sure how the paper reports accuracy if it times out). As the code for the baselines is not included in the supplementary material, I hope the authors can clarify these discrepancies. There are additional issues in the experimental section, e.g. there is no mention of hyperparameter tuning, c.f. the questions section.

Lastly, the performance of Dolphin is claimed to be state-of-the-art but I’ve seen several systems get better results on the considered benchmarks (the comparison is hard as actual numbers are not reported, and only bars). To give just some examples, Orvieto et al. (2023) report 94% for Pathfinder, and Manhaeve et al. (2021) report near-perfect accuracies for CLUTRR. I want to stress that I don’t think state-of-the-art results are necessary, but if they are claimed this should be properly supported.

In summary, the concerns about the novelty of the paper combined with the experimental evaluation issues unfortunately mean I cannot recommend acceptance.


**References**

Badreddine, S., Garcez, A. D. A., Serafini, L., & Spranger, M. (2022). Logic tensor networks. *Artificial Intelligence*.

Cohen, W., Yang, F., & Mazaitis, K. R. (2020). Tensorlog: A probabilistic database implemented using deep-learning infrastructure. *Journal of Artificial Intelligence Research*, *67*, 285-325.

Dang, M., Khosravi, P., Liang, Y., Vergari, A., & Van den Broeck, G. (2021). Juice: A julia
package for logic and probabilistic circuits. In *Proceedings of the AAAI Conference on Artificial Intelligence*.

Darwiche, A. (2020). An advance on variable elimination with applications to tensor-based computation. In *ECAI.*

Manhaeve, R., Dumančić, S., Kimmig, A., Demeester, T., & De Raedt, L. (2021). Neural probabilistic logic programming in DeepProbLog. *Artificial Intelligence*, *298*, 103504.

Marra, G., Giannini, F., Diligenti,  M., & Gori, M. (2019). Lyrics: A general interface layer to integrate logic inference and deep learning. In *Joint European Conference on Machine Learning and Knowledge Discovery in Databases*.

Orvieto, A., Smith, S. L., Gu, A., Fernando, A., Gulcehre, C., Pascanu, R., & De, S. (2023, July). Resurrecting recurrent neural networks for long sequences. In *International Conference on Machine Learning* (pp. 26670-26698). PMLR.

**Questions:**

- Line 213: “Tags are tensors that represent relative likelihoods”. Relative with respect to what? Relative likelihoods (as I understand it) represent a fraction between two likelihoods.
- The experimental section doesn’t really explain how the CLUTRR and Mugen tasks are solved by Dolphin. E.g. what are the Dolphin programs used here? I think it could be useful to at least include these in the Appendix.
- I found the naming of “Distribution” unclear. Unless I misunderstand it, a “Distribution” isn’t a probability distribution? (E.g. the Filter operation can remove probability mass.)
- How did you perform hyperparameter tuning? Did you tune the baselines in a similar fashion? Given that Table 2 compares total training time, better hyperparameters also affect the reported runtime.
- Why is there such a pronounced accuracy difference between Dolphin and Scallop in some experiments? From what I understand, the provenances like DAMP are essentially inherited from Scallop, so similar accuracy in Scallop should be possible (although not with the same runtime of course).

Minor comments:

- The paper mentions that an “NVIDIA GeForce RTX 4352” was used for the experiments for all experiments (besides CLUTRR). Is this a typo? I’m not aware of the existence of this specific model, and could not find anything about it on the internet. In contrast, the Appendix mentions that a GeForce RTX 2080 Ti was used.
- For Table 2, what is the unit of time? I assume this is in seconds, but I couldn’t find this anywhere.
- For Table 2, what is the provenance used for Dolphin? I assume this is DTKP-AM, but I couldn’t find this anywhere.
- Line 518: “these methods are point solutions”. What do you mean by "point solution"?
- The brackets on the citation on line 107 are wrong.
 - Figure 6 bottom would be more clear with a log y-scale.
- Several citations refer to the arXiv preprint instead of the conference publication (e.g. for neurASP, CLUTRR, and NeuralLog).

---

> ### Author Response · Authors · 2024-11-23
>
> We thank the reviewer for their insightful feedback and the extensive literature they brought to our attention. We will include a deeper discussion in the related work section on the systems from the literature and clarify other points within the revised manuscript.
>
> ## Novelty
>
> ### Tensor Operations for Neurosymbolic Learning.
>
> We thank the reviewer for pointing out the literature on parallelized neurosymbolic learning. We agree that the concept of using tensor operations for neurosymbolic learning is not new. However, systems such as LYRICS, Logic Tensor Networks (LTNs), and Tensorlog all have limited expressivity, which is one of the obstacles Dolphin aims to overcome. Specifically, they restrict the symbolic programs to first order logic and require the user to specify low-level information such as how variables are grounded and what their domains are. They also restrict the symbols to be in the form of tensors and the user-defined functions to consist of TensorFlow operations. These restrictions allow such systems to use TensorFlow to compile these programs into highly efficient computational graphs, but at the cost of expressivity. These frameworks also exclusively support simpler provenances and t-norms which are not sufficient for complex neurosymbolic programs. We describe these systems briefly in the related work section of the revised paper.
>
> On the other hand, Dolphin allows the user to track tags for specific symbols which can be arbitrary Pythonic objects. Dolphin programs further allow the user to manipulate Distributions over such symbols using arbitrarily complex code which may not necessarily translate to a computational graph. As such, there is a fine balance between the probabilistic computations, that happen over a GPU, and the symbolic computations, that take place on a CPU, all while maintaining a mapping between the two. This requirement sets a unique challenge addressed by Dolphin that we believe sets it apart from systems that use tensor operations for neurosymbolic learning. This fundamental design choice is also what allows Dolphin to be more expressive and flexible than existing systems. We also design Dolphin to be modular so that users can easily extend it to support new provenances and t-norms. As such, the t-norms used in LYRICS and LTN can be trivially added in a vectorized manner to Dolphin. We have added this discussion in the revised manuscript, along with a comparison of the expressivity of Dolphin with LTN for MNIST Sum-2, and demonstrate why HWF-N as written in Dolphin is not feasible in LTN. Please refer to Appendix E.
>
> ### Optimizing Probabilistic Computations via Tensors.
> Other works pointed by the reviewer, such as Dang et al. and Darwiche, focus solely on probabilistic computations rather than neurosymbolic frameworks. For instance, Juice by Dang et al. is a Julia package for logic and probabilistic circuits, which is not designed to be integrated with deep learning frameworks. On the other hand, Darwiche's work focuses on variable elimination with applications to optimize tensor-based computation. It will be interesting to see how Dolphin can be integrated with such systems to further improve the scalability and efficiency of neurosymbolic learning, and we will include a discussion on this in the revised manuscript. However, we still believe that Dolphin's novelty lies in its design that allows for the seamless integration of expressive neurosymbolic programs within deep learning frameworks, which is not addressed by the existing systems. We add this discussion in Appendix E.

---

> > ### Author Response · Authors · 2024-11-23
> >
> > ## Semantics
> >
> > ### Dolphin Primitive Semantics vs Provenance Semantics.
> >
> > We designed Dolphin to be a general-purpose neurosymbolic framework able to support various semantics, as long as they can be expressed as operations over tags tracked via the Distribution class. Dolphin assumes that the provenance supplied to it offers both the conjunction and disjunction operations that operate over combinations of tags from input symbols, as well as a way to translate tags to probabilities. As long as these assumptions are satisfied, the primitives of Dolphin preserve the semantics offered by the provenances. We add this in Appendix H.
> >
> > ### Top-K Semantics in DTKP-AM.
> >
> > It is true that the add-mult step in DTKP-AM is less precise than using WMC. However, this does not destroy the top-k semantics. For each symbol throughout the neurosymbolic program, we still track the top-k proofs that lead to the symbol and do not perform any addition operation over the proofs until we need to translate the tags into probabilities. This step only occurs when the `GetProbs` function is called, which in a typical neurosymbolic program is only called at the end, just before calculating the loss and performing backpropagation. This is also where WMC would get called as per the original paper proposing DTKP-WMC. We, therefore, believe that the add-mult operation in DTKP-AM is a reasonable vectorized approximation of WMC, and we even show that its performance is comparable to DTKP-WMC used in Scallop in the experiments. While this introduces clamping operations, PyTorch's implementation of clamp backpropagation ensures a gradient of 1 everywhere, even on the clamp boundaries (source: https://github.com/pytorch/pytorch/pull/7049).
> >
> > We include a detailed explanation of the DTKP provenance in the revised manuscript in Appendix C and a general discussion of these semantics and language choices in Appendix H.
> >
> > ## Language
> >
> > ### The Need for Primitives.
> >
> > We designed Dolphin to be integrated with deep learning frameworks like PyTorch to leverage their GPU acceleration capabilities. As such, we wanted the front-end to be as Pythonic as possible to enable deep learning practitioners to write neurosymbolic programs intuitively within their existing deep learning pipelines. Languages like Datalog, ASP, and ProbLog, are markedly different from Python, and follow a completely different paradigm. Since there are only 5 primitives introduced in Dolphin on top of the Distribution class, there is less of a barrier to writing complex neurosymbolic programs in an intuitive and Pythonic manner.
> >
> > In order to come up with the primitives, we studied several neurosymbolic tasks to determine the most common operations needed for these tasks. The primitives thus have parallels with vital Datalog operations, which we describe in more detail in Appendix H along with the motivation for introducing those primitives. As a result, Dolphin is as expressive as Scallop.
> >
> > ### ApplyIf vs (Apply + Filter).
> >
> > In Dolphin, operations within each primitive are executed independently of each other. In cases where we perform an `Apply` followed by `Filter`, this would require processing all possible combinations of symbols in the `Apply` operation, after which we can drop certain symbols via the `Filter` operation. Introducing `ApplyIf` allows for optimizations like preemptively dropping symbols violating the condition, which reduces the number of symbols that need to be processed. We find that the `ApplyIf` operation is required in enough cases to warrant its inclusion as a separate primitive in Dolphin.

---

> > > ### Author Response · Authors · 2024-11-23
> > >
> > > ## Experiments
> > >
> > > ### IndeCateR+ Results.
> > >
> > > We apologize for the discrepancies in the experimental results. We originally used the IndeCateR+ implementation provided by the ISED authors in their artifact, which was a CPU-centric implementation that was severely undersampled. We have rerun the MNIST and HWF experiments using the correct implementation and have updated the result tables as well as the text in the revised manuscript. Overall, Dolphin still outperforms IndeCateR+ in terms of accuracy and training time, but the gap is smaller. We also include the hyperparameters used for each experiment in the appendix A of the revised manuscript.
> > >
> > > ### Comparison with SOTA Tools.
> > >
> > > We wish to clarify that the performance of Dolphin is state-of-the-art among general-purpose neurosymbolic frameworks for benchmarks *except CLUTRR*, as was mentioned in the paper. The work in Orvieto et al. is specialized for long sequences, and was thus not included. We cite them and mention their performance in the revised manuscript in the Related Work section. We can also include a comparison with all versions of the PathFinder benchmark if the reviewers find it necessary.
> > >
> > > While DeepProbLog (Manhaeve et al., 2021) reports near-perfect accuracies for CLUTRR, they use negative mining techniques to provide additional supervision at train time. Scallop and Dolphin, on the other hand, stick to a traditional semi-supervised multiclass classification approach without producing additional labels. We cite their result in the experiment section of the revised manuscript.
> > >
> > > ## Questions
> > >
> > > **Line 213: “Tags are tensors that represent relative likelihoods”. Relative with respect to what? Relative likelihoods (as I understand it) represent a fraction between two likelihoods.**
> > >
> > > A. The tags represent the likelihoods for each symbol in a Distribution relative to other symbols in the same Distribution.
> > >
> > > **The experimental section doesn't really explain how the CLUTRR and Mugen tasks are solved by Dolphin. E.g. what are the Dolphin programs used here? I think it could be useful to at least include these in the Appendix.**
> > >
> > > A. We briefly describe the main program features needed to write the Dolphin programs for the CLUTRR and Mugen tasks in Appendix A, but include the programs themselves in the supplementary material. We will be happy to add all the programs to the appendix itself if the reviewer wishes.
> > >
> > > **I found the naming of “Distribution” unclear. Unless I misunderstand it, a “Distribution” isn't a probability distribution? (E.g. the Filter operation can remove probability mass.)**
> > >
> > > A. Yes, Distributions are not *probability distributions*, but simply a mapping from a set of symbols to their likelihoods. These likelihoods may be the logits from a neural model or results of Dolphin programs.
> > >
> > > **How did you perform hyperparameter tuning? Did you tune the baselines in a similar fashion? Given that Table 2 compares total training time, better hyperparameters also affect the reported runtime.**
> > >
> > > A. We mention the hyperparameters used in Appendix A. We do not tune the hyperparameters for the baselines, instead we use the default hyperparameters provided by the authors of the respective baselines. For Dolphin, we stick to the same batch size and top-k value as Scallop, but tune the learning rate such that the train accuracy upon convergence is maximized. Note that even though we report the total training time in Table 2, we primarily focus on the per epoch training time while determining the scaling factor for each benchmark.
> > >
> > > **Why is there such a pronounced accuracy difference between Dolphin and Scallop in some experiments? From what I understand, the provenances like DAMP are essentially inherited from Scallop, so similar accuracy in Scallop should be possible (although not with the same runtime of course).**
> > >
> > > A. This is primarily in the case of the Mugen task. For this task, we write the same program as Scallop’s in Dolphin and use the same base neural network. The only difference is in the backend neurosymbolic engine. We therefore hypothesize that Dolphin converges to a higher accuracy because it uses PyTorch for differentiating symbolic programs, while Scallop uses its own auto-differentiation framework.
> > >
> > > ## Minor Comments
> > >
> > > We will address the minor comments in the revised manuscript. The GPU used was indeed the RTX 2080 Ti, and we apologize for the typo. For Table 2, we refer to time in seconds. We mention the provenances used in Section 4.4. Point solutions mean the models are built for those specific applications.

---

> > > > ### Comment · Reviewer_1NAS · 2024-11-24
> > > >
> > > > **> (…) LYRICS, Logic Tensor Networks (LTNs), and Tensorlog all have limited expressivity, which is one of the obstacles Dolphin aims to overcome. Specifically, they restrict the symbolic programs to first order logic and require the user to specify low-level information such as how variables are grounded and what their domains are.**
> > > >
> > > > But Dolphin also reduces to a computational graph of $\oplus$ and $\otimes$ operations. This is precisely a logical circuit over some chosen algebra, i.e. the same as LTNs. In my opinion, you cannot start claiming things like improved expressivity without giving proper examples or proofs in the paper. Lastly, Dolphin also needs the specify how to ground symbols (perhaps it’s less verbose to do so, but still).
> > > >
> > > > **> They also restrict the symbols to be in the form of tensors and the user-defined functions to consist of TensorFlow operations. These restrictions allow such systems to use TensorFlow to compile these programs into highly efficient computational graphs, but at the cost of expressivity. These frameworks also exclusively support simpler provenances and t-norms which are not sufficient for complex neurosymbolic programs.**
> > > >
> > > > As the authors are surely aware, the operations in Dolphin are also tensor operations. (The fact that the grounding is different does not change anything about that.) So it’s puzzling to me how you can state that this is “at the cost of expressivity”. The formulation of real logic in logic tensor networks is very general, so I’d like to see what precise provenance of the Dolphin paper it couldn’t handle and why.
> > > >
> > > > **> As such, there is a fine balance between the probabilistic computations, that happen over a GPU, and the symbolic computations, that take place on a CPU, all while maintaining a mapping between the two. This requirement sets a unique challenge addressed by Dolphin that we believe sets it apart from systems that use tensor operations for neurosymbolic learning.**
> > > >
> > > > Grounding the computational graph on CPU and and running the computation on CPU is exactly what all the neurosymbolic frameworks I have mentioned do. I’m again puzzled how this is a “unique challenge addressed by Dolphin”. The way Dolphin constructs the computational graph is of course somewhat different, but if that’s the novelty of the paper it should be framed as such.
> > > >
> > > > **> We have added this discussion in the revised manuscript, along with a comparison of the expressivity of Dolphin with LTN for MNIST Sum-2**
> > > >
> > > > I thank the authors for giving a concrete example. However, to best of my knowledge this example only demonstrates that the Dolphin library is more intuitive to use compared to the LTN implementation, and does not demonstrate any difference in expressivity.
> > > >
> > > > **> (…) and demonstrate why HWF-N as written in Dolphin is not feasible in LTN.**
> > > >
> > > > Where is this demonstrated? I only found “Writing the same program in LTN is not feasible due to the requirement of concatenating strings and evaluating the expressions they represent”, which to the best of my understanding is false. HWF could very well be solved by LTNs.
> > > >
> > > > **> In order to come up with the primitives, we studied several neurosymbolic tasks to determine the most common operations needed for these tasks.**
> > > >
> > > > Ok, but shouldn’t this be described more in the paper how and why Dolphin covers (most of) the necessary common operations in relation with existing languages? Otherwise, the impression might be had that the design of Dolphin is overfitted to the 4 chosen experiments. I see this is covered a bit in Appendix H now, but it’s still rather superficial.
> > > >
> > > > **> As a result, Dolphin is as expressive as Scallop.**
> > > >
> > > > First you say that Dolphin is more expressive than first-order logic and now you say that it’s as expressive as Datalog, which famously cannot express many things from first-order logic. And again there is no proof to back up this claim (I checked appendix H).
> > > >
> > > > **> Introducing `ApplyIf` allows for optimizations like preemptively dropping symbols violating the condition, which reduces the number of symbols that need to be processed. We find that the `ApplyIf` operation is required in enough cases to warrant its inclusion as a separate primitive in Dolphin.**
> > > >
> > > > I understand that naively doing Apply followed by If is very inefficient. My point was that this could also have been optimized away by Dolphin, instead of relying on the user to do this manually.

---

> ### Comment · Reviewer_1NAS · 2024-11-24
>
> **> It is true that the add-mult step in DTKP-AM is less precise than using WMC. However, this does not destroy the top-k semantics.(…)**
>
> I get why the authors go for this approximation (it’s much easier than parallizing the actual probabilistic inference), but the paper should be clear about this. E.g. Appendix C states “We note that this approximation upper bounds the result from “full” WMC”, but really you have a upper bound on the top-k which is itself a lower bound on the full WMC. (i.e. there rema no guarantee at all). “We even hypothesize that in most cases, the add-mult approximation does not meaningfully affect the final result compared to full DTKP”; is this a different way to say Dolphin only considers tasks with mutually exclusive proofs (as e.g. Winters et al. targets)? Otherwise, there is considerable evidence in the literature that differentiating fuzzy semantics can be problematic (see e.g. van Krieken et al.).
>
> Winters, T., Marra, G., Manhaeve, R., & De Raedt, L. (2022). Deepstochlog: Neural stochastic logic programming. In *Proceedings of the AAAI Conference on Artificial Intelligence*.
>
> van Krieken, E., Acar, E., & van Harmelen, F. (2022). Analyzing differentiable fuzzy logic operators. *Artificial Intelligence*.
>
> **> We originally used the IndeCateR+ implementation provided by the ISED authors in their artifact, which was a CPU-centric implementation that was severely undersampled. We have rerun the MNIST and HWF experiments using the correct implementation and have updated the result tables as well as the text in the revised manuscript.**
>
> We thank the authors for clarifying this. I was somewhat surprised to hear that the ISED implementation of indecater would perform so much worse, as the reported accuracies in the ISED paper are much higher than what the Dolphin paper reported. If there was an error in the ISED paper, perhaps this should be communicated to the authors?
>
> **> The tags represent the likelihoods for each symbol in a Distribution relative to other symbols in the same Distribution.**
>
> But if this is a relative likelihood (as stated in the paper), what is the base likelihood you normalize with? Relative likelihood is a likelihood divided by another likelihood [1].
>
> [1]: https://en.wikipedia.org/wiki/Relative_likelihood
>
> **> Yes, Distributions are not *probability distributions*, but simply a mapping from a set of symbols to their likelihoods**
>
> So why call it a distribution if it’s not a distribution? And more importantly, w.r.t. what distribution is this likelihood of a symbol?
>
> More generally, I don’t understand why the paper and rebuttal talks about probabilities and likelihoods at several points while the authors at the same time also say that Dolphin does not have probabilistic semantics.
>
> **> We therefore hypothesize that Dolphin converges to a higher accuracy because it uses PyTorch for differentiating symbolic programs, while Scallop uses its own auto-differentiation framework.**
>
> Well the autodiff framework shouldn’t make any difference, unless the authors are implying that Scallop has errors in their autodiff implementation? This might be worth mentioning in the paper.
>
> I want to thank the authors for their extensive clarifications. However, I do not feel that my concerns about the paper are adequately addressed. I hence maintain my score.

---

> ### Author Response · Authors · 2024-11-25
>
> We thank the reviewer for taking the time to engage with us and helping us improve the presentation and clarify our contributions.
>
> ## About the Relationship of Dolphin with LTN
>
> We apologize for the confusion as we try to understand the relationship of Dolphin to LTN as it pertains to the concerns raised by the reviewer. We clarify our contributions with respect to the four design principles outlined in Section 3.1 using the simplest example, MNIST.
>
> ### The Dolphin Program for MNIST Sum-2
>
> We show the Dolphin program for MNIST Sum-2 here:
>
> ```python
> d1 = Distribution(model(img[0]), range(10))
> d2 = Distribution(model(img[1]), range(10))
>
> result_logits = GetProbs(Apply(d1, d2,  lambda x, y: x + y))
> ```
> There exist two kinds of computations in Dolphin: those occurring over symbols and those occurring over their corresponding probabilities. Symbols can be any objects (e.g. here the digits 0, 1, …, 9), and functions over them can be arbitrary operations (here the addition function).
>
> As we state in our first design principle, Dolphin allows for *flexible programmability*. To enable this, the symbolic computations (e.g. $f_\text{add}(1, 2)$) are run as sequential Python code on the CPU. This enables symbolic computations to be arbitrarily complex functions $f$ expressed in a high-level language like Python.
>
> To preserve this flexible programmability, Dolphin does not compile symbols and functions over them into PyTorch computation graphs. Doing so would require restricting the symbols to be tensors, and restricting the functions to be a chain of PyTorch operations over those tensors.
>
> On the other hand, Dolphin does compile the *probability computations* (e.g. d1(1) $\otimes$ d2(2)) over those symbols into PyTorch computation graphs that can be heavily parallelized, since the probabilities themselves are tensors on the GPU (assuming the model is run on the GPU), thus satisfying our second design principle of *end-to-end differentiability*.
>
> ### The LTN Program for MNIST Sum-2
>
> In contrast to Dolphin, LTN compiles *both* the symbol computations and the probability computations into TensorFlow computation graphs. The LTN program for MNIST Sum-2 is as follows:
>
> ```python
> ### Predicates
> Digit = ltn.Predicate.FromLogits(model, activation_function="softmax")
> ### Variables
> d1 = ltn.Variable("digits1", range(10))
> d2 = ltn.Variable("digits2", range(10))
> ### Operators
> Not = ltn.Wrapper_Connective(ltn.fuzzy_ops.Not_Std())
> And = ltn.Wrapper_Connective(ltn.fuzzy_ops.And_Prod())
> Or = ltn.Wrapper_Connective(ltn.fuzzy_ops.Or_ProbSum())
> Implies = ltn.Wrapper_Connective(ltn.fuzzy_ops.Implies_Reichenbach())
> Forall = ltn.Wrapper_Quantifier(ltn.fuzzy_ops.Aggreg_pMeanError(),semantics="forall")
> Exists = ltn.Wrapper_Quantifier(ltn.fuzzy_ops.Aggreg_pMean(),semantics="exists")
>
>
> # mask
> add = ltn.Function.Lambda(lambda inputs: inputs[0]+inputs[1])
> equals = ltn.Predicate.Lambda(lambda inputs: inputs[0] == inputs[1])
>
> ### Axioms
> @tf.function
> def axioms(images_x, images_y, labels_z, p_schedule=tf.constant(2.)):
> 	images_x = ltn.Variable("x", images_x)
> 	images_y = ltn.Variable("y", images_y)
> 	labels_z = ltn.Variable("z", labels_z)
> 	axiom = Forall(
>         	ltn.diag(images_x,images_y,labels_z),
>         	Exists(
>             	(d1,d2),
>             	And(Digit([images_x,d1]),Digit([images_y,d2])),
>             	mask=equals([add([d1,d2]), labels_z]),
>             	p=p_schedule
>         	),
>         	p=2
>     	)
> 	result_logits = axiom.tensor
> 	return result_logits
> ```
>
> This does not satisfy our first design principle due to the reasons mentioned above: constants in LTN programs have to be grounded as tensors rather than remaining arbitrary Python objects, and the functions have to be compilable into a TensorFlow computation subgraph. Note that user-defined functions need to be supplied using `ltn.Function.Lambda` or `ltn.Predicate.Lambda`, which can only accept expressions over tensors rather than Python functions over arbitrary objects.
>
> This is the fundamental difference between Dolphin and systems like LTN and Scallop. On one hand, similar to LTN, Dolphin uses tensor computations and GPU support to enhance scalability compared to Scallop. On the other hand, as the reviewer also noted, Dolphin is more intuitive, allowing programmers to write symbolic computations over dynamic data structures in a high-level language like Python.

---

> ### Author Response · Authors · 2024-11-25
>
> ### Scaling in the Presence of Sequential Symbol Computations
>
> Given that we do not compile symbol computations into PyTorch computation graphs to satisfy our first design principle, satisfying the third design principle (scalability) becomes a challenge. We address this by condensing symbols in the Distribution class into a single collection stored in CPU RAM while maintaining tags as a GPU tensor ($b \times N \times T$, where $b$ is the batch size, $N$ is the number of symbols, and $T$ is the shape of the tag) as described in our response to Reviewer mznt. This results in there being one set of CPU-based computations for the entire batch of samples rather than one set of computations for each sample within the Distribution, which is typical of other neurosymbolic frameworks. This allows Dolphin to maintain the benefits of parallelism even while the user-defined functions are executed sequentially.
>
> We demonstrate this by sharing the breakdown of the time taken for symbol computations and tag computations within the forward pass of the Dolphin program for the HWF task that we showed in the response to Reviewer mznt. The first row shows the time taken during the forward pass when the Dolphin program is run sequentially on the CPU with no parallelism. The second row shows the time taken when tag computations are parallelized on the GPU over batches of 64 samples each. The times annotated with C and G indicate time spent on the CPU and GPU, respectively:
>
> | Config                    	| Time for UDF (s) | Time for Tag Computations (s) | Total Time (s) |
> |-------------------------------|------------------|-------------------------------|----------------|
> | No Parallelism            	| 36.24 (C)    	| 461.02 (C)                	| 497.26     	|
> | Parallelized Tag Computations | 14.13 (C)    	| 75.125 (G)                	| 89.25      	|
>
> This also means that due to Dolphin’s design, increases in batch size result in fewer total CPU operations over the entire training epoch, since the set of CPU operations is shared for the entire batch while parallelizing more tag computations over the entire batch. We observe similar behavior for MNIST Prod-N (Table 4 in Appendix F), where as we increase the batch size, the training time reduces for large values of N.
>
> ### Addressing Tunability
>
> These design choices also serve to satisfy the last design principle of tunability: since Dolphin decouples the symbol computations from the probability computations, we are able to treat provenances as another hyperparameter in the deep learning pipeline, and even allow developers to add more provenances without changing the Dolphin framework itself.
>
> ### HWF-N in LTN
>
> We do not claim that one cannot write HWF using LTN. What we intend to say is that to write the same HWF program as Dolphin in LTN, LTN needs to support strings as constants and allow programmers to supply arbitrary Python functions like `eval`. As discussed earlier, LTN does not allow either. So in order to solve HWF, one must write a much less intuitive and more complex program.
>
> ## About the Choice of Primitives
>
> We thank the reviewer for their feedback on Appendix H, and we will add more details in a revised manuscript. Is there something specific the reviewer would like to see in those details?
>
> That said, the fact that one can supplement these primitives with user-defined functions means that these primitives can be easily used to write programs for other tasks.
>
> ## About Expressivity in terms of Scallop
>
> We say that we are as expressive as Scallop because Scallop offers more features than Datalog, such as foreign functions and algebraic data types.
>
> ## About ApplyIf
>
> Adding optimizations to the computation graph generated by Dolphin is indeed an interesting area of research. We leave this to future work since it is orthogonal to the design of a general-purpose neurosymbolic framework.
>
> ## About DTKP-AM
>
> We do specify in the revised paper that DTKP-AM is an approximation of DTKP-WMC. We also do not claim to provide any guarantees, and have clarified our appendix to say that DTKP-AM only upper bounds DTKP-WMC. We test our both DAMP and DTKP-AM on the following example that does not involve mutually exclusive proofs:
>
> ```python
> a = Distribution(self.mnist_net(a_imgs), range(10))
> res = a + a + a + a
> ```
>
> Using DTKP-AM results in a 98.86% accuracy, while using DAMP results in a 90.51% accuracy. While this is a simple example, it does show that DTKP-AM does provide benefits when the proofs in the symbolic computation are not mutually exclusive.

---

> ### Author Response · Authors · 2024-11-25
>
> ## About the likelihoods
>
> We remove the “relative” term from the current revision for describing the likelihoods, since we do not normalize the likelihoods within Distributions. Typically, any normalization would occur in a neural network via functions like the sigmoid or softmax functions. But we do not enforce any normalization explicitly within the Dolphin program or the Distributions themselves. We agree that calling them Distributions may be confusing and incorrectly imply that they are specifically probability distributions. We will change the name to something less confusing if the reviewer suggests it.
>
> ## About the Scallop results
>
> We have not seen any behavior to indicate that there is a bug in Scallop’s backend. It is more likely the case that PyTorch performs optimizations that result in a computation graph that is easier to converge over.
>
> ## About the IndeCateR+ results
>
> We did notify the authors of the ISED paper, and they have updated their numbers as well.

---

### Official Review · Reviewer_fUaG · 2024-10-27

**Soundness:** 3
**Presentation:** 2
**Contribution:** 2
**Rating:** 5
**Confidence:** 3

**Summary:**

This work presents Dolphin, a brand new framework designed to enhance the scalability of neurosymbolic learning. Dolphin allows developers to write differentiable symbolic programs in Python, utilizing PyTorch for end-to-end GPU acceleration. The framework conveys flexible programmability, end-to-end differentiability, scalability, and tunability, which are essential to handling complex programs and large datasets effectively. Experimental results demonstrate that Dolphin significantly outperforms existing frameworks.

**Strengths:**

- A new framework for neurosymbolic learning, namely Dolphin, is developed.
- Dolphin shows superior performance compared to existing frameworks.

**Weaknesses:**

My major concern is that the paper is not well structured and hard to follow. In the introduction section, the authors criticized existing frameworks that they must use a separate CPU-based backend and suffer from the slow inter-process data transfers. For me, this is implementation-specific, and it does not drive me to the reasons why we should redesign the entire framework. Although the authors further discuss the challenges in lines 52-67, I find it rather irrelevant to the aforementioned limitation of slow inter-process data transfers.

Moreover, as a new framework, there lacks an overview to depict the layered structures. This prevents readers from having a general picture. It is hard to tell why the designs/implementations could realize the core principles. Additionally, I cannot map the core principles to the challenges discussed in the introduction, either.

**Questions:**

- It would be better to provide a breakdown of the training time (e.g., according to Figure 1) to justify the major efficiency improvement of Dolphin.
- In Figure 5, for small tasks that all competitors could converge within the time limit, why Dolphin has a better accuracy?
- In Figure 6, there is a trade-off between accuracy and training time for different provenances (DAMP vs. DTKP-AM). How should we select the most suitable one in practice?

---

> ### Author Response · Authors · 2024-11-23
>
> We thank the reviewer for their feedback and will incorporate it to make the paper more readable and easily understandable. We address the reviewer’s concerns below:
>
> ## About the challenges outlined in the paper
>
> Using a separate CPU-based backend is not merely an implementation choice but a fundamental design limitation in most neurosymbolic frameworks. As mentioned in the introduction, these frameworks must perform symbolic computations while tracking tags and probabilities across input, intermediate, and output symbols, which inherently involve complex, variable-sized data structures and operations that are difficult to parallelize.
>
> This limitation forces existing frameworks to rely on CPU backends. Since deep learning models are typically implemented in Python, frameworks face a trade-off: either implement symbolic reasoning directly in Python, which is slow for CPU-based operations or use a separate backend, typically written in some compiled language. While the latter is faster, it requires transferring data structures, operations, logits, and gradients between the neural pipeline and the external backend, introducing inter-process latency. We clarify the reason of interprocess data transfers in the introduction.
>
> ## About the core principles
>
> Dolphin tries to address the following core principles as follows:
> * Flexible programmability: The Distribution abstraction, along with the associated primitives, allow for expressing complex neurosymbolic programs over arbitrary Python objects through user-defined Python functions.
> * End-to-end differentiability on GPUs: The Distribution abstraction maintains a mapping from symbols to tags stored as PyTorch tensors on GPUs sourced directly from neural network models. Furthermore, the provenances governing the tag operations are defined in a differentiable manner, allowing Dolphin to harness PyTorch’s GPU support and auto-differentiation mechanisms.
> * Scalability: Operations over Distributions defined through its primitives can be batched easily, which are processed in a vectorized manner, allowing Dolphin programs to scale with larger and more complex datasets.
> * Tunability: The modular design of Distributions, where the primitives are defined independent of the provenances, allows users to rapidly plug in and test out different provenances to select the one best suited for their task.
>
> Together, these core principles directly address the challenges of both problem complexity as well as data complexity while inhibiting the need for a separate CPU-based backend. Principles 1 and 4 address the issue of program complexity, while principles 2 and 3 focus on scaling with data complexity. We describe how these principles map to the challenges in the revised manuscript in Section 3.1.
>
> ## Other Questions
>
> ### Why does Dolphin have a better accuracy?
> We attribute the difference in accuracies between Dolphin and ISED / IndeCateR+ to the underlying design of each baseline. IndeCateR+ and ISED are sampling-based gradient approximation methods, which are inherently stochastic and may not converge to the optimal solution. As for Scallop, this only happens in Mugen, where we write the same program as Scallop’s in Dolphin and use the same base neural network. The only difference here is in the backend neurosymbolic engine. We therefore hypothesize that Dolphin converges to a higher accuracy because it uses PyTorch for differentiating symbolic programs, thus benefitting from Python’s optimizations over the computational graph, while Scallop uses its own auto-differentiation framework.
>
> ### How should we select the most suitable provenance in practice?
> The choice of provenance depends on many factors, including the complexity of the program, the independence of variables within the program, and the desired trade-off between accuracy and training time. Typically, one would try each provenance and use the one yielding the best accuracy. If both provenances perform similar, it is more practical to choose DAMP over DTKP due to its efficiency.
>
> ### About the breakdown of training times.
> Unfortunately, finding the time taken for the inter-process data transfer requires profiling systems like Scallop in detail, which is not readily available.

---

> > ### Comment · Reviewer_fUaG · 2024-11-25
> >
> > Thanks for the detailed explanation. I wish the authors could make the logic chain clearer, such as the foundamental source of the limitations that existing works suffer from, the techniques that your work proposes (not simply the goals you wish to achieve), and why the proposed techniques are able to address the limitations. Currently, the manuscript focuses on introducing the degisn of your work, but readers may not understand the rationale behind your design.

---

> > > ### Author Response · Authors · 2024-11-25
> > >
> > > We thank the reviewer for taking the time to engage with us. Their feedback is vital for improving the paper. We clarify our contributions with respect to the four design principles outlined in Section 3.1 using the simplest example, MNIST.
> > >
> > > #### The Dolphin Program for MNIST Sum-2
> > >
> > > We show the Dolphin program for MNIST Sum-2 here:
> > >
> > > ```python
> > > d1 = Distribution(model(img[0]), range(10))
> > > d2 = Distribution(model(img[1]), range(10))
> > >
> > > result_logits = GetProbs(Apply(d1, d2,  lambda x, y: x + y))
> > > ```
> > > There exist two kinds of computations in Dolphin: those occurring over symbols and those occurring over their corresponding probabilities. Symbols can be any objects (e.g. here the digits 0, 1, …, 9), and functions over them can be arbitrary operations (here the addition function).
> > >
> > > As we state in our first design principle, Dolphin allows for *flexible programmability*. To enable this, the symbolic computations (e.g. $f_\text{add}(1, 2)$) are run as sequential Python code on the CPU. This enables symbolic computations to be arbitrarily complex functions $f$ expressed in a high-level language like Python.
> > >
> > > To preserve this flexible programmability, Dolphin does not compile symbols and functions over them into PyTorch computation graphs. Doing so would require restricting the symbols to be tensors, and restricting the functions to be a chain of PyTorch operations over those tensors.
> > >
> > > On the other hand, Dolphin does compile the *probability computations* (e.g. d1(1) $\otimes$ d2(2)) over those symbols into PyTorch computation graphs that can be heavily parallelized, since the probabilities themselves are tensors on the GPU (assuming the model is run on the GPU), thus satisfying our second design principle of *end-to-end differentiability*.
> > >
> > > #### The LTN Program for MNIST Sum-2
> > >
> > > In contrast to Dolphin, LTN compiles *both* the symbol computations and the probability computations into TensorFlow computation graphs. The LTN program for MNIST Sum-2 is as follows:
> > >
> > > ```python
> > > ### Predicates
> > > Digit = ltn.Predicate.FromLogits(model, activation_function="softmax")
> > > ### Variables
> > > d1 = ltn.Variable("digits1", range(10))
> > > d2 = ltn.Variable("digits2", range(10))
> > > ### Operators
> > > Not = ltn.Wrapper_Connective(ltn.fuzzy_ops.Not_Std())
> > > And = ltn.Wrapper_Connective(ltn.fuzzy_ops.And_Prod())
> > > Or = ltn.Wrapper_Connective(ltn.fuzzy_ops.Or_ProbSum())
> > > Implies = ltn.Wrapper_Connective(ltn.fuzzy_ops.Implies_Reichenbach())
> > > Forall = ltn.Wrapper_Quantifier(ltn.fuzzy_ops.Aggreg_pMeanError(),semantics="forall")
> > > Exists = ltn.Wrapper_Quantifier(ltn.fuzzy_ops.Aggreg_pMean(),semantics="exists")
> > >
> > >
> > > # mask
> > > add = ltn.Function.Lambda(lambda inputs: inputs[0]+inputs[1])
> > > equals = ltn.Predicate.Lambda(lambda inputs: inputs[0] == inputs[1])
> > >
> > > ### Axioms
> > > @tf.function
> > > def axioms(images_x, images_y, labels_z, p_schedule=tf.constant(2.)):
> > > 	images_x = ltn.Variable("x", images_x)
> > > 	images_y = ltn.Variable("y", images_y)
> > > 	labels_z = ltn.Variable("z", labels_z)
> > > 	axiom = Forall(
> > >         	ltn.diag(images_x,images_y,labels_z),
> > >         	Exists(
> > >             	(d1,d2),
> > >             	And(Digit([images_x,d1]),Digit([images_y,d2])),
> > >             	mask=equals([add([d1,d2]), labels_z]),
> > >             	p=p_schedule
> > >         	),
> > >         	p=2
> > >     	)
> > > 	result_logits = axiom.tensor
> > > 	return result_logits
> > > ```
> > >
> > > This does not satisfy our first design principle due to the reasons mentioned above: constants in LTN programs have to be grounded as tensors rather than remaining arbitrary Python objects, and the functions have to be compilable into a TensorFlow computation subgraph. Note that user-defined functions need to be supplied using `ltn.Function.Lambda` or `ltn.Predicate.Lambda`, which can only accept expressions over tensors rather than Python functions over arbitrary objects.
> > >
> > > This is the fundamental difference between Dolphin and systems like LTN and Scallop. On one hand, similar to LTN, Dolphin uses tensor computations and GPU support to enhance scalability compared to Scallop. On the other hand, as the reviewer also noted, Dolphin is more intuitive, allowing programmers to write symbolic computations over dynamic data structures in a high-level language like Python.

---

> ### Author Response · Authors · 2024-11-25
>
> #### Scaling in the Presence of Sequential Symbol Computations
>
> Given that we do not compile symbol computations into PyTorch computation graphs to satisfy our first design principle, satisfying the third design principle (scalability) becomes a challenge. We address this by condensing symbols in the Distribution class into a single collection stored in CPU RAM while maintaining tags as a GPU tensor ($b \times N \times T$, where $b$ is the batch size, $N$ is the number of symbols, and $T$ is the shape of the tag) as described in our response to Reviewer mznt. This results in there being one set of CPU-based computations for the entire batch of samples rather than one set of computations for each sample within the Distribution, which is typical of other neurosymbolic frameworks. This allows Dolphin to maintain the benefits of parallelism even while the user-defined functions are executed sequentially.
>
> We demonstrate this by sharing the breakdown of the time taken for symbol computations and tag computations within the forward pass of the Dolphin program for the HWF task that we showed in the response to Reviewer mznt. The first row shows the time taken during the forward pass when the Dolphin program is run sequentially on the CPU with no parallelism. The second row shows the time taken when tag computations are parallelized on the GPU over batches of 64 samples each. The times annotated with C and G indicate time spent on the CPU and GPU, respectively:
>
> | Config                    	| Time for UDF (s) | Time for Tag Computations (s) | Total Time (s) |
> |-------------------------------|------------------|-------------------------------|----------------|
> | No Parallelism            	| 36.24 (C)    	| 461.02 (C)                	| 497.26     	|
> | Parallelized Tag Computations | 14.13 (C)    	| 75.125 (G)                	| 89.25      	|
>
> This also means that due to Dolphin’s design, increases in batch size result in fewer total CPU operations over the entire training epoch, since the set of CPU operations is shared for the entire batch while parallelizing more tag computations over the entire batch. We observe similar behavior for MNIST Prod-N (Table 4 in Appendix F), where as we increase the batch size, the training time reduces for large values of N.
>
> #### Addressing Tunability
>
> These design choices also serve to satisfy the last design principle of tunability: since Dolphin decouples the symbol computations from the probability computations, we are able to treat provenances as another hyperparameter in the deep learning pipeline, and even allow developers to add more provenances without changing the Dolphin framework itself.

---

### Official Review · Reviewer_mznt · 2024-11-01

**Soundness:** 3
**Presentation:** 3
**Contribution:** 2
**Rating:** 8
**Confidence:** 4

**Summary:**

The authors introduce Dolphin, which is a pytorch-friendly framework for performing neuro-symbolic computations.  The neural component of a computation is assumed to be a model, such as an MNIST digit classifier, which outputs a discrete set of symbols (e.g the digits 0-9), with a probability attached to each of them.  The symbolic component is a python program which runs a computation over the symbols.  The result of symbolic execution is a pytorch tensor, representing final output probabilities for each possible result, which is end-to-end differentiable with the neural components.  The authors apply Dolphin to several of neuro-symbolic benchmarks, and show that it is faster than competing frameworks.

Dolphin essentially works by running the symbolic program for every possible combination of input symbols, and tracking the probability of each combination.  The symbolic program is executed on CPU, but Dolphin evaluation will merge different traces of the program which have the same output into batches.  The probabilities can then be computed using batch operations that are GPU-friendly, as well as being end-to-end differentiable with pytorch.

The authors also provide two different mechanisms, which they call provenances, for tracking probabilities.  The DAMP provenance tracks all probabilities, while DTKP tracks only the top-K proofs.

**Strengths:**

The paper is very well written, and describes the basic execution and batching mechanism clearly, at least for the DAMP provenance.  The Dolphin framework does seem to be an improvement over SOTA in terms of basic usability for certain classes of neuro-symbolic programs.

**Weaknesses:**

The authors spend a lot of time talking about the easy parts of the problem, and fail to adequately discuss the hard parts.  As a result, they gloss over two glaring weaknesses that I see with using this approach to solve anything other than cherry-picked trivial problems.

The first issue is the combinatorics.  When evaluating a function f(A,B), where A and B are distributions over symbols, evaluation must evaluate f(a,b) for every possible combination of symbols { a | a \in A }, and { b | b\ in B }.  Depending on the exact problem, this can easily lead to an combinatorial explosion in the number of possible outputs.  The authors test their code on the "sum of MNIST digits" problem, where the combinatorics are reasonable; even given 20 digits, there are at most 181 possible answers.  If they were to instead try the "product of MNIST digits", which is a tiny change to the code, then the number of possible outputs would balloon, and the technique would likely fail.

The second issue is control flow.  As a symbolic computation, the "sum of digits" has no loops or branches, and thus is trivially easy to batch.  The authors mention that they support recursive computations, but those generally require a branch to terminate the recursion, and often have divergent control flow.  In the presence of branches, different traces of the program take different paths, and no longer cleanly batch together.

The usual solution (e.g. in CUDA programs) is that when evaluation encounters a branch, it splits the batch of traces into a then-branch and an else-branch, and then merges the traces together again afterwards.  Without merging, the traces will continue to diverge on subsequent branches, until each trace is operating independently at batch size 1, and the benefits of parallelism are lost.

Merges happen at the join points in a control-flow graph, which requires the underlying library to build a control-flow graph.  Alternatively, since there are only two batched operations (conjunction and disjunction), the authors could first construct an (unbatched) DAG of operations, and then merge/batch together independent nodes of the DAG after the fact, in the style of Looks et al. "Deep learning with dynamic computation graphs," or Neubig et al. "On-the-fly operation batching in dynamic computation graphs."

However, the authors make no mention of any machinery to analyze control-flow, build control-flow graphs, or otherwise auto-batch in the presence of divergent control flow.  In fact, they do not even provide a discussion or examples of how to write recursive computations with their library at all, despite claiming that it is possible.

My main objection with both of these issues is that the authors simply don't discuss these problems at all, when IMO they are very clearly major limitations that affect the kind of programs that Dolphin is able to run.

A further weakness of the writing itself is that the authors do not do a good job of explaining the DTKP provenance, which seems like it's quite important.  I have several criticisms here.  First, it is possible that choosing only the top-K proofs after each operation will address the combinatorics issue, which would be a big deal.  However, I'm uncertain, because the authors gloss over combinatorics problem altogether without discussion.  Second, the authors claim that their mechanism for merging DTKP tags is equivalent to weighted model counting, but this claim is wholly unsubstantiated.  I didn't really understand the formula in Table 1 at all, including how infinities get into the tags.  At the very least, the authors should provide a detailed discussion of DKTP in the appendix, ideally with a proof of how it relates to WMC, if space within the paper itself is an issue.

Finally, the authors mention that wrt. to the HWF task, "the DTKP-AM provenance is more effective than DAMP since the tags in DAMP provenance lack the structure needed to capture the semantics of the symbolic program."  This statement seems important, but really requires further explanation; I don't understand it at all.  Providing HWF as a worked example (perhaps in the appendix) would be valuable to anybody who actually wants to use Dolphin.

Errors:

Line 310: "Its conjuction operation is defined as the addition of probabilities, and its disjunction is defined as the multiplication of probabilities."  Unless my understanding is way off base, shouldn't this be the other way around?  For independent observations, p(A and B) means multiplying p(A) and p(B)?  That's what the authors show in Table 1.

**Questions:**

Please see "weaknesses" above -- in particular my confusion with the DTKP provenance.

---

> ### Author Response · Authors · 2024-11-23
>
> ## About the potential of combinatorial explosions in computations
>
> ### Handling Combinatorial Explosions.
>
> We thank the reviewer for highlighting the issue of combinatorial explosion. This is indeed a fundamental challenge in neurosymbolic programs as a whole. Dolphin mitigates this by leveraging the Distribution class, which condenses symbols into a single collection stored in CPU RAM while maintaining tags as a GPU tensor (b x N x T, where b is the batch size, N is the number of symbols, and T is the shape of the tag). As shown in Figure 2(b), this approach reduces symbolic overhead by avoiding redundant evaluations for each sample in a batch, unlike frameworks like Scallop, where each sample is independently evaluated. While tag evaluations still involve all combinations across all samples in a batch, they are computed in a vectorized manner on the GPU. We include this explanation in Appendix F.
>
> ### Results for MNIST Product-N.
>
> We conducted a preliminary experiment for the MNIST Product-N benchmark suggested by the reviewer. We summarize the results below. For each N, we report the time taken per epoch in seconds averaged over 5 epochs as well as the accuracy achieved:
>
> | N  | Time per Epoch (s) | Accuracy |
> |----|---------|----------|
> | 4  | 11.42          	| 0.96 	|
> | 8  | 12.55          	| 0.95 	|
> | 16 | 27.45          	| 0.94 	|
> | 20 | 36.59          	| 0.92 	|
>
> We get further scalability improvements by increasing the batch size. Increasing the batch size from 64 to 128 yields these numbers:
>
> | N  | Time per Epoch (s) | Accuracy |
> |----|---------|----------|
> | 4  | 8.92          	| 0.97 	|
> | 8  | 9.15          	| 0.95 	|
> | 16 | 15.71          	| 0.89 	|
> | 20 | 18.73          	| 0.85 	|
>
> We see the effect of combinatorial explosion as the time taken per epoch increases with N. However, the explosion does not render the computation infeasible, and Dolphin is still able to achieve high accuracy. The runtime also scales within reason, and increasing the batch size reduces the runtime due to batched computations within Dolphin. We include these results in Appendix F.
>
> ## How does Dolphin deal with control flow?
>
> ### Control Flow in Dolphin.
>
> In Dolphin, control flow largely exists within the lambda functions supplied to the `Apply`, `ApplyIf`, and `Filter` operations, which can be arbitrary Python functions over the symbols in the Distributions. As discussed in Section 3.2.2, these functions can include complex operations like if-then-else branches, loops, and even recursion. The nature of these functions means that they cannot be parallelized over the GPU. Instead, they are executed sequentially on the CPU, while the associated tags are computed parallely on the GPU. We optimize the design of the Distribution class so that there is one set of CPU-based computations for the entire batch of samples rather than one set of computations for each sample, which is typical of other neurosymbolic frameworks. This allows Dolphin to maintain the benefits of parallelism even while the user-defined functions are executed sequentially. We include this discussion in Appendix D.
>
> ### Control Flow for HWF.
>
> We demonstrate how Dolphin handles control flow by showing the time taken for the HWF task split by the time spent on the CPU and GPU. The code for this task is shown in Appendix G, and involves complex control flows within its user-defined functions (UDFs) like Python’s `eval` operation. The first row shows the time taken during the forward pass when the Dolphin program is run sequentially on the CPU with no parallelism. The second row shows the time taken when tag computations are parallelized on the GPU over batches of 64 samples each. The times annotated with C and G indicate time spent on the CPU and GPU, respectively:
>
> | Config | Time for UDF (s) | Time for Tag Computations (s) | Total Time (s) |
> |---|---|---|---|
> | No Parallelism  | 36.24 (C) | 461.02 (C) | 497.26  |
> | Parallelized Tag Computations | 14.13 (C) | 75.125 (G)  | 89.25 |
>
> Observe that the time, both for UDF computation and for Tag computation, decreases as we move from sequential CPU evaluation to the batched evaluation. Due to Dolphin’s design, increases in batch size result in fewer total CPU operations over the entire training epoch, since the set of CPU operations is shared for the entire batch, while parallelizing more tag computations over the entire batch. We include these results in Appendix D as well.
>
> ### Recursion.
>
> In order to write recursive computations in Dolphin, one has two choices: either supply a recursive user-defined function to the Dolphin primitives, or write a more fine-grained program in Python that uses Dolphin primitives in the base case as well as the recursive case, set to terminate once a condition is met. Here, the diverging control flows can be merged using the `Union` primitive. We discuss recursion and control flow further in Appendix D and show an example on writing recursive programs using Dolphin.

---

> > ### Author Response · Authors · 2024-11-23
> >
> > ## Questions about DTKP-AM
> > We add an appendix that explains in detail how DTKP-AM works. We also wish to clarify that the add-mult step in DTKP-AM is not equivalent to WMC but is rather a vectorized approximation, albeit less precise than full-fledged WMC. We discuss this in Appendix C.
> >
> > ### About the infinities.
> > Because tensors are rectangular and must have uniform dimensions, there are cases where there exist varying numbers of proofs (less than K) involving varying numbers of input symbols. As such, we require a way to denote the absence of an input symbol from a proof as well as the absence of a proof itself from a tag. Specifically, an absent symbol should not influence the probability of a proof (obtained by multiplying the probabilities of present symbols) and an absent proof should not influence the probability of a tag (obtained by adding the probabilities of individual proofs).
> >
> > We realize this by choosing to represent the absence of such symbols using $\+infty$ and absent proofs by vectors of $\-infty$. Using the “norm” function, absent symbols are thus denoted as 1, allowing us to multiply the tag tensor along the column dimension to obtain probabilities of proofs. Again using the same function, absent proofs are denoted by 0, allowing us to sum the proof probabilities.
> >
> > ### Why does it work better for HWF?
> >
> > The structure of the HWF program is as follows: A neural network classifies each image into one of 14 symbols (digits 0–9 and operators +, -, *, /). Each symbol is stored as a string (e.g., “0”, “1”, “+”), and concatenation operations applied to Distributions over such symbols yield a final Distribution ∗D∗ over expressions. During concatenation, partial parsing adds complexity via a user-defined function. Finally, each expression is evaluated to produce a Distribution over numeric results. The implementation and details are provided in Appendix G.
> >
> > For such a complex Dolphin program, using a simple provenance like DAMP proves insufficient for longer sequences since the tags of all possible combinations of symbols are collated into a single number. On the other hand, DTKP-AM is able to track the top-k proofs for each symbol, pruning out the less probable proofs. Furthermore, since each proof is a collection of input symbols leading to a specific output, once the loss is calculated, gradients can be backpropagated directly to the input symbols that had the most influence on the output. On the other hand, the gradients may be distributed across all symbols in DAMP as it backpropogates through each intermediate computation regardless of their role in the computation of the output, resulting in slower convergence. We include this explanation in Appendix G as well.

---

> > > ### Author Response · Authors · 2024-11-24
> > >
> > > We also noticed an inconsistency in Table 1 and updated the definitions of $\textbf{0}$ and $\textbf{1}$ for the DTKP-AM provenance to fix this issue. Now the definitions are consistent: $\textbf{0}$ denotes the absence of a proof from a tag, while $\textbf{1}$ denotes the absence of symbols within an individual proof.

---

> > ### Comment · Reviewer_mznt · 2024-11-26
> >
> > Thank you for the clarification, and for all of the detailed information you added to the appendix.  Color-coding the additional text in blue was especially convenient.  I believe that most of my concerns have been addressed, and I am raising my score.
> >
> > Unfortunately, I tend to agree with reviewer 1NAS that the material here is somewhat hard to follow, especially since so much information is now split between the main paper and the appendix, and the appendix itself is somewhat hastily organized.  I would encourage the authors to try to clarify the presentation before the camera-ready copy.
> >
> > In particular, since this paper will likely serve as an entry point for researchers who might be inclined to use Dolphin, the following point should be made clear within the main text of the paper:
> >
> > There are actually two ways in which recursion and control flow can potentially be used in a Dolphin program.  The first mechanism is inside a function $f$ in Apply($f$), because $f$ may contain arbitrary python code.  (That's the simple case.)  The second mechanism is in the outer loop which manipulates distributions.  Because distributions are sets of symbols, control flow and/or recursion in the outer loop is restricted to set-operations, such as tests for set size and equality.  Seeing the worked example for compute_path in Figure 9 was very helpful for me to understand how to structure a Dolphin program in this second case; there is no divergent control flow in the outer loop.  Ideally, that example (which also helps to explain the practical use of the ApplyIf and Union operators) should be in the text of the main paper -- I would have been much less confused if I had seen it to start with.
> >
> > Wrt. to combinatorics, the combinatorics for MNIST product are not as bad as I initially thought when I wrote my earlier review -- the symbol count is still only 53362 for N=20, which is manageable.  I still think this issue warrants further discussion, especially since the authors admit that is a "fundamental challenge in neurosymbolic programs as a whole."  A more general search routine, which preferentially explores the most likely paths, and prunes unlikely symbols from the distribution (e.g. MCTS) seems like a critical direction for future research, IMO.

---

> > > ### Author Response · Authors · 2024-11-27
> > >
> > > We thank the reviewer for their feedback and detailed comments to help us improve the clarity and presentation of the paper. We have revised the submission, which now contains a section on control flows and recursion (Section 3.3), including the transitive closure example (Figure 5). We are committed to improving the clarity of the paper and the appendix, and we will add the other discussions suggested by the reviewers as well.

---

### Official Review · Reviewer_rSem · 2024-11-03

**Soundness:** 3
**Presentation:** 3
**Contribution:** 3
**Rating:** 8
**Confidence:** 3

**Summary:**

The paper proposes DOLPHIN, a scalable neurosymbolic learning framework that integrates symbolic reasoning efficiently within deep learning models. Unlike existing frameworks that struggle with CPU-GPU data transfer bottlenecks, DOLPHIN enables symbolic programs to operate as a set of inheritanted PyTorch nn.module, allowing efficient GPU-based differentiation.

**Strengths:**

1. The authors propose an end-to-end neurosymbolic framework that makes all progress differentiable.
2. With a glance at the provided code, the DOLPHIN is a lightweight implementation of the framework integrated with PyTorch, having a potential opportunity to support the neurosymbolic community.
3.  The evaluation on 13 benchmarks across 5 neurosymbolic tasks show the advantage of the proposed DOLPHIN.

**Weaknesses:**

1. It seems DOLPHIN only supports neurosymbolic programs with deterministic symbolic processes. For example, if HWF task requires the neural part to predict both numbers and operators (+,-,*,/), the symbolic part cannot be programmed with the Apply function. How DOLPHIN deal with this situation?

**Questions:**

Please refer to weaknesses 1.

Other questions:

1. What is the limitation of the proposed DOLPHIN?
2. Are lambda functions fast enough? Do we require doing some acceleration for the proposed operations, such as designing some specific CUDA kernel or triton functions?

---

> ### Author Response · Authors · 2024-11-23
>
> ## About the HWF task
>
> We thank the reviewer for pointing out that Dolphin currently only supports deterministic symbolic processes. Before we address this, we would like to explain how the HWF example as suggested by the reviewer can be implemented in Dolphin. In the HWF task presented in the paper, the neural model does predict both numbers and operators by classifying each image into 14 classes: 10 digits (0-9) and 4 operators (+, -, *, /). The symbolic program then computes the result of the expression represented by the image. Since Dolphin allows for arbitrary Python objects and functions to be used as symbols and operations, we can easily represent the HWF task as follows:
>
> ```python
> symbols = [ str(i) for i in range(10) ] + [ '+', '-', '*', '/' ]
> res = Distribution(model(img[0]), symbols)
>
> for i in range(1, expr_length):
> 	op = Distribution(model(img[i]), [ '+', '-', '*', '/' ])
> 	res = apply(lambda x, y: x + y, res, op)
>
> res = apply(lambda expr: eval(expr), res)
> result_logits = get_probabilities(res)
> ```
>
> Here, the Distribution associates the logits of the neural model with strings representing the digits and operators. The `apply` function is used to concatenate these symbols to form strings representing entire expressions. The final result is then evaluated using the `eval` function. Note that this is a naive version of the HWF task, and the full code for the HWF model is shown in Appendix G.
>
> However, currently, Dolphin does not support non-deterministic symbolic processes. This includes cases where the symbolic program itself may not be known and may need to be approximated. This could be addressed in many ways, such as by supporting weighted operations where the weights themselves are learned during backpropagation, or by even using LLMs to generate the symbolic program. We leave this to future work but will include this as a limitation in the revised manuscript.
>
> ## Limitations
>
> We mention a limitation of Dolphin in Section 6, namely that programs in Dolphin need to be written in a batched manner, which may pose a challenge for users without experience in deep learning.
>
> ## Lambda functions
>
> The lambda functions in Dolphin can be any Python function. As a result, the efficiency of the functions itself is dependent on the user. While Dolphin does not provide any acceleration for the lambda functions themselves, the operations on the tags stored in Distributions are already optimized for GPU acceleration. In general, user-defined functions do not pose a significant bottleneck in our benchmarks. To demonstrate this, we present a breakdown of the time required for the lambda functions (referred to as UDFs) and the time required for computing tags in Appendix D.

---

> > ### Comment · Reviewer_rSem · 2024-11-24
> > **Response to authors**
> >
> > Thank you for your reply. My concerns are almost solved. I will keep my score.

---

### Author Response · Authors · 2024-11-23
**Summary**

We thank the reviewers for their insightful suggestions and feedback. We have responded to each reviewer individually. Please let us know if there are any questions before the end of the discussion period.

We have revised our submission based on the feedback. All changes are highlighted in blue. We summarize the main changes:
1. We provide a detailed description of the DTKP-AM provenance in Appendix C.
2. We describe how control flow and recursive computations are specified in Dolphin in Appendix D.
3. We compare Dolphin with other tensor-based neurosymbolic techniques in Appendix E and the related work section.
4. We discuss combinatorial explosions in Appendix F.
5. We include the full neurosymbolic model for the HWF task in Appendix G and explain why DTKP-AM performs better.
6. We describe the motivation for the Dolphin language and discuss the choice of primitives along with the semantics supported in Appendix H.
7. We correct issues with IndeCateR+ in the experiments and cite SOTA techniques for Path and CLUTRR.
8. We provide additional clarifications throughout the paper where needed.

---

### Author Response · Authors · 2024-12-03

As the rebuttal period concludes, we summarize the key improvements and discussions regarding our submission.

Reviewers rSem, mznt, and fUaG highlighted the importance of scalable neurosymbolic frameworks and commended Dolphin's integration of symbolic reasoning with PyTorch for improved scalability and usability. In response to requests for clarification on control flow, recursion, and DTKP-AM semantics, we provided detailed explanations and incorporated them into the manuscript.

To address reviewer 1NAS’s concerns about novelty, we included systems like LTN, LYRICS, and Tensorlog in the related work section. We clarified that Dolphin supports arbitrary Python functions for symbolic computations, unlike frameworks like LTN that relied on TensorFlow operations. Examples such as MNIST Sum-2 and HWF-N demonstrated Dolphin’s flexibility and intuitive programmability while maintaining differentiability and scalability. Regarding DTKP-AM, we clarified it as an approximation of DTKP-WMC, balancing scalability and precision. Terminology updates included removing "relative" likelihood references and revising "Distribution" for clarity.

To address reviewer mznt's concerns, we expanded discussions on combinatorial explosions and control flows, illustrated with a Dolphin code snippet for transitive closure. We also included results from MNIST Product-N to show Dolphin's scalability. Optimizations in Dolphin's batch processing allow efficient handling of recursive tasks and complex computations, as detailed in the revised manuscript.

Lastly, we corrected experimental results for IndeCateR+, ensured fair comparisons, and informed the ISED authors of these updates. We hope our rebuttal has addressed all concerns and welcome any final comments.

---

### Meta-Review · Area_Chair_ohJi · 2024-12-21

**Metareview:**

The paper introduces Dolphin, a framework designed to enhance the scalability of neurosymbolic learning by integrating symbolic reasoning into deep learning models using PyTorch. Dolphin allows symbolic programs to be written as PyTorch modules, enabling end-to-end differentiation on GPUs. The reviewers are divided over this paper. On one hand, two reviewers believe that this is a new approach as mentioned in the abstract, while two more reviewers feel this is more nuanced case. The latter reviewers express that there is relevant work in neurosymbolic methods that are tensorized and those are not included in the baselines. This is a major drawback, since that would not place Dolphin appropriately in the literature and would significantly reduce the claimed novelty.

This is a challenging case: the paper can add value to the community, but only once compared with appropriate frameworks and benchmarks, especially on more challenging cases where the brute force approach might not scale. Thus, at the moment, this is a borderline reject, based on the reviewers' comments.

**Additional Comments On Reviewer Discussion:**

The reviewers raised various concerns regarding the novelty and the relationship to previous papers. In addition, reviewers point out that the claim on improved expressivity should be more rigorously determined.

---

### Decision · Program_Chairs · 2025-01-22

Reject